



**Methane emissions in the United States, Canada, and Mexico: Evaluation of national methane emission inventories and sectoral trends by inverse analysis of in situ (GLOBALVIEWplus CH4 ObsPack) and satellite (GOSAT) atmospheric observations**

Xiao Lu[1,2], Daniel J. Jacob[2], Haolin Wang[1], Joannes D. Maasakkers[3], Yuzhong Zhang[2,4,5], Tia R. Scarpelli[2], Lu Shen[2], Zhen Qu[2], Melissa P. Sulprizio[2], Hannah Nesser[2], A. Anthony Bloom[6], Shuang Ma[6], John R. Worden[6], Shaojia Fan[1], Robert J. Parker[7,8], Hartmut Boesch[7,8], Ritesh Gautam[9], Deborah Gordon[10,11], Michael D. Moran[12], Frances Reuland[13], Claudia A Octaviano Villasana[14]

1 School of Atmospheric Sciences, Sun Yat-sen University, Zhuhai, Guangdong Province, China
2 Harvard John A. Paulson School of Engineering and Applied Sciences, Harvard University, Cambridge, MA, USA
3 SRON Netherlands Institute for Space Research, Utrecht, the Netherlands
4 School of Engineering, Westlake University, Hangzhou, Zhejiang Province, China
5 Institute of Advanced Technology, Westlake Institute for Advanced Study, Hangzhou, Zhejiang Province, China
6 Jet Propulsion Laboratory, California Institute of Technology, Pasadena, CA, USA
7 National Centre for Earth Observation, University of Leicester, Leicester, UK
8 Earth Observation Science, Department of Physics and Astronomy, University of Leicester, Leicester, UK
9 Environmental Defense Fund, Washington, DC, USA
10 RMI, New York, NY, USA;
11 Watson Institute for International and Public Affairs, Brown University, Providence, RI, USA
12 Environment and Climate Change Canada, Toronto, ON, Canada
13 RMI, Boulder, CO, USA
14 Instituto Nacional de Ecología y Cambio Climático (INECC), Mexico City, Mexico

*Correspondence to:* Xiao Lu (luxiao25@mail.sysu.edu.cn)





**Abstract**

We quantify methane emissions and their 2010-2017 trends by sector in the contiguous United States (CONUS), Canada, and Mexico by inverse analysis of in situ (GLOBALVIEWplus $CH_4$ ObsPack) and satellite (GOSAT) atmospheric methane observations. The inversion uses as prior estimate the national anthropogenic emission inventories for the three countries reported by the US Environmental Protection Agency (EPA), Environment and Climate Change Canada (ECCC), and the Instituto Nacional de Ecologia y Cambio Climatico (INECC) in Mexico to the United Nations Framework Convention on Climate Change (UNFCCC), and thus serves as an evaluation of these inventories in terms of their magnitudes and trends. Emissions are optimized with a Gaussian mixture model (GMM) at 0.5°×0.625° resolution and for individual years. Optimization is done analytically using log-normal error forms. This yields closed-form statistics of error estimates and information content on the posterior (optimized) estimates, allows better representation of the high tail of the emission distribution, and enables construction of a large ensemble of inverse solutions using different observations and assumptions. We find that GOSAT and in situ observations are largely consistent and complementary in the optimization of methane emissions for North America. Mean 2010-2017 anthropogenic emissions from our base GOSAT + in situ inversion, with ranges from the inversion ensemble, are 36.9 (32.5-37.8) Tg a$^{-1}$ for CONUS, 5.3 (3.6-5.7) Tg a$^{-1}$ for Canada, and 6.0 (4.7-6.1) Tg a$^{-1}$ for Mexico. These are higher than the most recent reported national inventories of 26.0 Tg a$^{-1}$ for the US (EPA), 4.0 Tg a$^{-1}$ for Canada (ECCC), and 5.0 Tg a$^{-1}$ for Mexico (INECC). The correction in all three countries is largely driven by a factor of 2 underestimate in emissions from the oil sector with major contributions from the south-central US, western Canada, and southeast Mexico. Total CONUS anthropogenic emissions in our inversion peak in 2014, in contrast to the EPA report of a steady decreasing trend over 2010-2017. This reflects combined effects of increases in emissions from the oil and landfill sectors, decrease from the gas, and flat emissions from the livestock and coal sectors. We find decreasing trends in Canadian and Mexican anthropogenic methane emissions over the 2010-2017 period, mainly driven by oil and gas emissions. Our best estimates of mean 2010-2017 wetland emissions are 8.4 (6.4-10.6) Tg a$^{-1}$ for CONUS, 9.9 (7.8-12.0) Tg a$^{-1}$ for Canada, and 0.6 (0.4-0.6) Tg a$^{-1}$ for Mexico. Wetland emissions in CONUS show an increasing trend of 2.6 (1.7-3.8) % a$^{-1}$ over 2010-2017 correlated with precipitation.





## 1. Introduction

Atmospheric methane (CH$_4$) is the most important anthropogenic greenhouse gas after carbon dioxide (CO$_2$). Natural emissions are mainly from wetlands. Anthropogenic emissions are from many sectors including the oil/gas supply chain, coal mining, livestock, and waste management. Individual countries must report their anthropogenic methane emissions by sector to the United Nations in accordance with the United Nations Framework Convention on Climate Change (UNFCCC, 1992). These national emission inventories are mainly constructed by bottom-up methods as the product of activity data and emission factors, following methodological guidelines from the Intergovernmental Panel on Climate Change (IPCC). The emission factors are highly variable and have large uncertainties, leading to errors in estimating national emissions, their trends, and the contributions of different sectors (Kirschke et al., 2013; Saunois et al., 2020). Top-down methods involving inversion of atmospheric methane observations can usefully diagnose these errors (Houweling et al., 2017). Here, we use an inverse analysis of 2010-2017 in situ and satellite observations of atmospheric methane over North America to evaluate national emission inventories and their trends by sector for the United States (US), Canada, and Mexico.

US anthropogenic methane emissions are reported yearly by the US Environmental Protection Agency (EPA, 2021) as part of the Inventory of US Greenhouse Gas Emissions and Sinks (GHGI). Methane emissions for the year 2012, from the 2016 version of this inventory (EPA, 2016), were spatially allocated on a 0.1º × 0.1º (10 × 10 km) grid by Maasakkers et al. (2016) to enable its evaluation using top-down methods. Results using analysis of atmospheric methane measurements from ground, aircraft, and satellite platforms show larger methane emissions than reported in the GHGI, particularly for the oil/gas industry (Alvarez et al., 2018; Zhang et al., 2020; Lu et al., 2021; Maasakkers et al. 2021; Qu et al., 2021) and for livestock (Lu et al., 2021; Yu et al., 2021). Atmospheric observations also suggest an increasing trend of US anthropogenic emissions over the past decade (Turner et al., 2015; Sheng et al., 2018a; Lan et al., 2019; Maasakkers et al., 2021), while the GHGI indicates a decrease (EPA, 2021).

Anthropogenic methane emissions for Canada are reported yearly by Environment and Climate Change Canada (ECCC) (ECCC, 2020a; 2021) as part of the National Inventory Report (NIR). Atmospheric observations again indicate an underestimate of emissions from oil/gas production (Atherton et al., 2017; Johnson et al., 2017; Chan et al., 2020; Baray et al., 2021; Lu et al., 2021; Tyner and Johnson, 2021) but a decrease of these emissions over the past decade (Lu et al., 2021; Maasakkers et al., 2021). Scarpelli et al. (2021) recently allocated the 2020 version of the ECCC NIR for the year 2018 (ECCC 2020a) on a 0.1º × 0.1º grid and our work is the first to use it in an inverse analysis.

Mexico's anthropogenic methane emissions are reported by the Instituto Nacional de Ecología y Cambio Climático (INECC) in Mexico's National Inventory of Greenhouse Gases and Compounds (INEGyCEI) for selected years (INECC and SEMARNAT, 2018). The last communication to the UNFCCC was in 2015 and this inventory was allocated to a 0.1º×0.1º grid by Scarpelli et al. (2020). A recent inverse analysis of satellite data finds oil/gas emissions to be underestimated by a factor of 2 over eastern Mexico (Shen et al., 2021).



The above top-down studies except for Baray et al. (2020) and Lu et al. (2021) used either in situ or satellite observations but not both. Satellite observations have better data coverage but are less sensitive to emissions and have larger uncertainties, particularly at high latitudes. In a previous inverse analysis (Lu et al., 2021), we showed that in situ and satellite observations provide complementary global information for inverse analyses of methane emissions. That inversion was conducted at 4°×5° resolution, too coarse for specific evaluation of national inventories.

Here we apply extensive in situ observations from surface sites, towers, ships, and aircraft (GLOBALVIEWplus $CH_4$ ObsPack data compilation) together with the Greenhouse Gases Observing Satellite (GOSAT) observations, in an inverse analysis for 2010-2017 to optimize methane emissions and their year-to-year variability at up to 0.5×0.625° resolution for North America. We use as prior estimates the gridded national emission inventories from EPA (US), ECCC (Canada), and INECC (Mexico), so that our results can inform inventory improvement planning at the emission sector level. Following Lu et al. (2021), we use an analytical inversion method that provides closed-form characterization of error statistics and information content of the inverse solution, and also allows us to compare quantitatively the information from the in situ and satellite observations.

## 2. Methods

We use methane observations from the GLOBALVIEWplus $CH_4$ ObsPack in situ data (Section 2.1) and/or GOSAT satellite retrievals (Section 2.2) with the GEOS-Chem chemical transport model (Section 2.4) as the forward model, to optimize a state vector of mean methane emissions for individual years (Section 2.3) covering the North American continent at a spatial resolution of up to 0.5°×0.625°. We derive posterior estimates of the state vector and the associated error covariance matrix by analytical solution to the Bayesian optimization problem (Section 2.5). Our base inversion uses GOSAT + in situ observations and our best choices of inversion parameters. We also present results from an ensemble of sensitivity inversions using observation subsets (in situ or GOSAT) and varying inversion parameter assumptions (e.g. different error distributions). We attribute inversion results to different methane emission sectors with the methodology described in Section 2.6.

### 2.1 In situ methane observations

We use the comprehensive database of in situ (surface, tower, shipboard, and aircraft) methane observations over North America for 2010-2017 from the GLOBALVIEWplus $CH_4$ ObsPack v1.0 product compiled by the National Oceanic and Atmospheric Administration (NOAA) Global Monitoring Laboratory (Cooperative Global Atmospheric Data Integration Project, 2019). Following Lu et al. (2021), data from surface and tower sites are sampled only during daytime (10-16 local time) and averaged as daytime mean values on individual days for use in the inversion. For sites with standard deviations larger than 30 ppb, we exclude data points that depart by more than two standard deviations from the mean because such conditions are difficult to simulate with the transport model. For other sites we exclude data points that depart by more than three standard deviations from the mean. We also exclude aircraft measurements higher than 9 km above sea level as these measurements would have weak sensitivity to surface fluxes.



The in situ observations thus include 49742 data points from surface sites, 15285 from towers, 56 from ship cruises, and 26620 from aircraft campaigns over North America and adjacent waters (Figure 1a). The number of available in situ observations per year increases from 10830 in 2010 to 13593 in 2017. All these in situ data points are used in the base inversion to optimize methane emissions for individual years. We also conduct sensitivity inversions by only using surface and tower sites with continuous eight-year records for trend analyses.

## 2.2 GOSAT satellite methane observations

The GOSAT satellite launched in 2009 measures the backscattered solar radiation from a sun-synchronous orbit at around 13:00 local time (Kuze et al., 2016). Methane is retrieved in the 1.65 µm shortwave infrared absorption band. We use the column-averaged dry-air methane mixing ratios from the University of Leicester version 9.0 Proxy $XCH_4$ retrieval (Parker et al., 2020a). Comparison with ground-based methane observations from the Total Carbon Column Observing Network (TCCON) shows that the retrieval has a single-observation precision of 13 ppb and an overall global bias of 9 ppb that is removed from the Proxy $XCH_4$ data (Parker et al., 2020a). Here we use a total of 205875 (25734 per year on average) GOSAT retrievals for 2010-2017 over North America in the inversion, excluding glint data over the oceans and data poleward of 60º which are not representatively sampled and for which errors are large (Figure 1b).

## 2.3 Prior emission inventories

We use as prior estimates of anthropogenic methane emissions the gridded versions of the official national inventories for the US (EPA, 2016), Canada (ECCC, 2020), and Mexico (INECC and SEMARNAT, 2018) (Maasakkers et al., 2016; Scarpelli et al., 2020, 2021). These emissions are listed in Table 1 for individual countries and the spatial distributions for major sectors are shown in Figure 2. We assume no year-to-year trend in the prior emissions, so that trends from the inversion are solely driven by observations. Prior anthropogenic emissions for the contiguous US (CONUS) are 28.7 Tg a$^{-1}$. Anthropogenic US emissions outside CONUS (mostly Alaska, not optimized in the inversion) account for only 0.3 Tg a$^{-1}$ according to Maasakkers et al. (2016). The latest GHGI report from EPA (2021) gives mean emissions of 26.0 Tg a$^{-1}$ for 2010-2017. Prior anthropogenic emissions for Canada are 3.7 Tg a$^{-1}$. The most recent 2021 version of the ECCC NIR gives a mean of 4.0 Tg a$^{-1}$ for 2010-2017 (ECCC, 2021). Mexico anthropogenic emissions are 5.0 Tg a$^{-1}$. 2015 is the latest available year from INECC.

Prior methane emissions from wetlands are the 0.5º × 0.5º gridded mean monthly values for 2010-2017 from the nine highest-performance members of the WetCHARTs v1.3.1 inventory ensemble (Ma et al, 2021), selected for their fit to the global GOSAT inversion results of Zhang et al. (2021). This choice of prior estimate effectively corrects the large overestimates of wetland emissions for North America previously found in inversions of GOSAT and aircraft data when using the overall mean of the WetCHARTs v1.0 ensemble (Sheng et al., 2018b; Maasakkers et al., 2021). We do not include interannual variability from WetCHARTs because it is highly uncertain and we prefer to have it informed by the observations. Unlike in our global inversion (Lu et al., 2021), we do not optimize the relative seasonal



variation of wetland emissions and instead have it imposed by the prior estimate (Parker et al., 2020b). Prior estimates of open fire emissions are the daily values for individual years from the Global Fire Emissions Database (GFED) version 4s (van der Werf et al., 2017). Other small natural emissions (seepages, termites) are as described in Lu et al. (2021).

### 2.4 The GEOS-Chem forward model

We use the nested version of the GEOS-Chem 12.5.0 chemical transport model (http://geos-chem.org, last access: 6 April 2021) (Wecht et al., 2014) as the forward model for the inversion. The model is driven by MERRA-2 reanalysis meteorological fields at their native $0.5° \times 0.625°$ resolution (Gelaro et al., 2017). Methane loss from atmospheric oxidation is as described in Lu et al. (2021) but is inconsequential here because it is negligibly slow compared to the timescale for ventilation of the North American domain. Soil uptake of methane is from the MeMo model v1.0 (Murguia-Flores et al., 2018) but is very small and therefore not optimized in the inversion.

The GEOS-Chem model simulation is conducted at $0.5° \times 0.625°$ resolution over the North America domain (130-55°W, 15-65°N) (Fig.1) for the 2010-2017 period, with initial and dynamic boundary conditions archived every 3 hours from a global 2010-2017 simulation at $4° \times 5°$ resolution using methane emissions and sinks previously optimized with GOSAT observations (Lu et al., 2021). This means that GOSAT observations over North America are used twice, once for the global inversion (along with other observations worldwide) and once for the North American inversion, but this is inconsequential because the sole purpose of the global optimization is to avoid biases in boundary conditions that would cause spurious corrections to emissions within the inversion domain (Wecht et al., 2014). Lu et al. (2021) show that their optimized simulation is unbiased in comparison to global zonal mean observations for 2010-2017 but we still find some residual bias for individual years. We therefore optimize the mean boundary conditions for individual years on each side of the domain (north, south, west, east) as part of the North American inversion.

### 2.5 Inversion procedure

Our state vector $x$ to be optimized in the inversion includes spatially resolved emissions in North America and boundary conditions for each year of 2010-2017. Although we could technically optimize methane emissions for each of the $0.5° \times 0.625°$ native model grid elements, the observations do not have sufficient coverage to constrain emissions everywhere at that resolution and doing so would introduce large smoothing errors in the inversion (Wecht et al., 2014). Following Turner and Jacob (2015) and Maasakkers et al. (2021), we use instead a Gaussian mixture model (GMM) to determine the emission patterns that can be constrained effectively by the inversion. This is done by projecting the native-resolution methane emissions onto 600 Gaussian functions optimized to fit the location, magnitude, and distribution of methane emissions for different sectors as given by the prior estimates. Optimal construction of the GMM aggregates regions with weak or homogeneous emissions while preserving native resolution for strong localized source regions. The Gaussian functions overlap, providing additional high-resolution structure in the inverse solution on the $0.5° \times 0.625°$ native grid. The state vector $x$ for individual years is defined as the emission of each of the 600 Gaussians, plus the correction to the





model boundary conditions as described earlier, for a total dimension $n = 604$.


The inversion finds the optimal estimate of $\boldsymbol{x}$ by minimizing the Bayesian cost function $J(\boldsymbol{x})$ (Brasseur and Jacob, 2017):

$$J(\boldsymbol{x}) = (\boldsymbol{x} - \boldsymbol{x_A})^T \boldsymbol{S_A}^{-1}(\boldsymbol{x} - \boldsymbol{x_A}) + \gamma(\boldsymbol{y} - \boldsymbol{F}(\boldsymbol{x})^T \boldsymbol{S_O}^{-1}(\boldsymbol{y} - \boldsymbol{F}(\boldsymbol{x}))\ (1),$$

where $\boldsymbol{x_A}$ is the prior estimate of $\boldsymbol{x}$, $\boldsymbol{S_A}$ denotes the prior error covariance matrix, $\boldsymbol{y}$ is the observation
vector, $\boldsymbol{S_O}$ denotes the observation error covariance matrix, $\gamma$ is a regularization factor (see below), and $\boldsymbol{F}(\boldsymbol{x})$ represents the GEOS-Chem simulation of $\boldsymbol{y}$. The GEOS-Chem forward model $\boldsymbol{F}(\boldsymbol{x})$ as implemented here is strictly linear (because methane sinks are not optimized), so that the model can expressed as $\boldsymbol{y} = \boldsymbol{Kx} + \boldsymbol{c}$, where $\boldsymbol{K} = \partial \boldsymbol{y}/\partial \boldsymbol{x}$ represents the Jacobian matrix and $\boldsymbol{c}$ is a constant. Minimizing the cost function (Eq.1) by solving $\nabla_{\boldsymbol{x}} J(\boldsymbol{x}) = \boldsymbol{0}$ yields closed-form posterior estimates of
the state vector $\widehat{\boldsymbol{x}}$, its error covariance matrix $\widehat{\boldsymbol{S}}$, and the averaging kernel matrix $\boldsymbol{A}$ (Rodgers, 2000; Brasseur and Jacob, 2017):

$$\widehat{\boldsymbol{x}} = \boldsymbol{x_A} + \boldsymbol{G}(\boldsymbol{y} - \boldsymbol{Kx_A})\ (2),$$
$$\widehat{\boldsymbol{S}} = (\gamma \boldsymbol{K}^T \boldsymbol{S_O}^{-1} \boldsymbol{K} + \boldsymbol{S_A}^{-1})^{-1}\ (3),$$
$$\boldsymbol{A} = \frac{\partial \widehat{\boldsymbol{x}}}{\partial \boldsymbol{x}} = \boldsymbol{I_n} - \widehat{\boldsymbol{S}}\boldsymbol{S_A}^{-1}\ (4),$$

where $\boldsymbol{G}$ in Eq.2 is the gain matrix,

$$\boldsymbol{G} = \frac{\partial \widehat{\boldsymbol{x}}}{\partial \boldsymbol{y}} = (\gamma \boldsymbol{K}^T \boldsymbol{S_O}^{-1} \boldsymbol{K} + \boldsymbol{S_A}^{-1})^{-1} \gamma \boldsymbol{K}^T \boldsymbol{S_O}^{-1}\ (5).$$

The averaging kernel matrix $\boldsymbol{A}$ in Eq. 4 quantifies the sensitivity of the posterior estimate to changes in the true value, and therefore measures the information content provided by the observing system for correcting the prior estimates and returning the true values as posterior estimates. We refer to the diagonal
elements of $\boldsymbol{A}$ as the averaging kernel sensitivities, and to the trace of $\boldsymbol{A}$ as the degrees of freedom for signal (DOFS), representing the number of pieces of independent information on the state vector obtained from the observing system (Rodgers, 2000). Our inversion returns the posterior estimates of mean emissions and averaging kernel sensitivities for each Gaussian, and these can be mapped additively to the $0.5°×0.625°$ grid using their spatial distributions on the grid.


Analytical solution to equation (2), and inference of error statistics and information content from equations (3)-(4), requires explicit construction of the Jacobian matrix $\boldsymbol{K}$. We construct $\boldsymbol{K}$ by conducting GEOS-Chem simulations where each element of the state vector is perturbed separately. This is readily done computationally as an embarrassingly parallel problem. Analytical solution has several
advantages relative to the more widely used variational (numerical) approach. (1) It identifies the true minimum in the cost function. (2) It provides complete explicit forms of the posterior error covariance and averaging kernel matrices. (3) It enables a range of sensitivity analyses at no significant computational cost modifying the inversion parameters and adding/subtracting observations.

To construct the prior error covariance matrix $\boldsymbol{S_A}$, we assume a 50% error standard deviation for individual Gaussians in the base inversion (and we test the sensitivity to that assumption as will be described later), with no spatial error covariance so that $\boldsymbol{S_A}$ is diagonal. There is necessarily some spatial





covariance in the prior estimates since the Gaussians have spatial overlap, and there is also some spatial covariance in the forward model error contributing to $S_O$, but these are difficult to quantify. The former would underestimate the information content of the observations while the latter would overestimate it. We effectively correct for this using the regularization parameter γ as described below, and we further rely on our inversion ensemble rather than the posterior error covariance matrix to characterize the error in our posterior solution.

The standard assumption of Gaussian error statistics in the cost function of equation (1) is required to achieve an analytical solution but may lead to unphysical negative emissions (Miller et al., 2014) and fail to capture the heavy tail of the emission distribution (Zavala-Araiza et al., 2015; Frankenberg et al., 2016; Alvarez et al., 2018). We solve this problem by optimizing for $\ln(x)$ instead of $x$, with the error on ln $(x)$ following a normal Gaussian distribution, i.e., lognormal errors for $x$ (Maasakkers et al., 2019). The forward model is then nonlinear, so that the solution must be solved iteratively with a transformed Jacobian matrix $K'_N = \partial y / \partial \ln(x)$ at each iteration $N$. Once the original Jacobian matrix $K = \partial y / \partial x$ for the linear model has been computed, we can derive $K'_N$ immediately at any iteration by $\partial y_i / \partial \ln(x_j) = x_j \partial y_i / \partial x_j$, where $i$ and $j$ represent the indices of the observation and state vector elements, respectively. The iterative solution is obtained with the Levenberg–Marquardt method (Rodgers, 2000) for each iteration $N$:

$$x'_{N+1} = x'_N + \left(\gamma K'_N{}^T S_O{}^{-1} K'_N + (1+\kappa) S'_A{}^{-1}\right)^{-1} \left((\gamma K'_N{}^T S_O{}^{-1}(y - Kx_N) - S'_A{}^{-1}(x'_N - x'_A)\right)$$

(6),

where $x' = \ln(x)$ with the initial value $x'_0$ from the prior estimate, and $\kappa = 10$ is a coefficient for the iterative approach to the solution (Rodgers, 2000). $S'_A$ (with diagonal elements denoted by $s'_A$) is the prior error covariance matrix for the inversion in log space, and can be derived from the original prior error covariance matrix $S_A$ (with diagonal elements denoted by $s_A$) following (Maasakkers et al., 2019):

$$s'_A = \left(\frac{\left(\ln\left(\frac{x_A + \sqrt{s_A}}{x_A}\right) + \left|\ln\left(\frac{x_A - \sqrt{s_A}}{x_A}\right)\right|\right)}{2}\right)^2 \quad (7).$$

We adopt as convergence criterion that the maximum difference between $x'_{N+1}$ and $x'_N$ elements be smaller than 5‰, at which point we adopt $\widehat{x'} = x'_{N+1}$ as our posterior solution. The posterior error covariance and averaging kernel matrices $\widehat{S'}$ and $A'$ on the log solution are obtained by replacing $S_A$ and $K$ with $S'_A$ and $K'$ in Eqs. (3) and (4). Optimization of emissions in log space means that $\widehat{x'}$ is a best estimate of the median of the log-normal error distribution rather than the mean. The mean values for spatial and sectoral aggregation purposes can be inferred from the properties of the lognormal distribution as $x_{j(mean)} = x_{j(median)} e^{\widehat{s'}_{jj}/2}$ where $\widehat{s'}_{jj}$ is the corresponding diagonal element of the posterior error covariance matrix in log space, i.e., the geometric error standard deviation. The boundary conditions are still optimized with normal error distributions, assuming an error standard deviation of 10 ppb.





The above describes our base inversion. We also conduct sensitivity inversions using different error assumptions. This includes 1) using the quadrature sum of error variances for all sectors contributing to a given Gaussian with a cap of 50% following Maasakkers et al. (2021), resulting in a 43% error on average; 2) to 4) using the normal error distributions (then with the linear Jacobian matrix) with 50%, 95%, and the quadrature sum of errors for individual Gaussians as error variances; 5) assuming an error

standard deviation of 5 ppb for boundary conditions.

The observation error covariance matrix $S_O$ includes contributions from measurement and forward model errors. We compute it following the residual error method originally described by Heald et al. (2004) and previously used by Lu et al. (2021). A GEOS-Chem simulation with prior emission estimates yields a prior model estimate $F(x_A)$ of concentrations at the observation points. The mean 2010-2017

discrepancy between the observations and the prior model, $\overline{y - F(x_A)}$, is determined for each grid cell (for GOSAT), individual observation site (surface and tower), and observation platform (shipboard and aircraft). $\overline{y - F(x_A)}$ is taken to represent the systematic bias in the prior emissions to be corrected in the inversion. The residual term, $\varepsilon_O = y - F(x_A) - \overline{y - F(x_A)}$, represents the random observation error

including contributions from the measurements, the forward model, and the representation of the observation points on the model grid (Heald et al., 2004). The variance of $\varepsilon_O$ provides the diagonal terms of $S_O$. The resulting observation error standard deviations average 13 ppb for GOSAT, 26 ppb for surface sites, 39 ppb for towers, 19 ppb for ships, and 22 ppb for aircraft. The observation error is larger for in situ than for satellite observations, even though the in situ measurements are more precise, because the

forward model error is larger for local points than for atmospheric columns (Turner et al., 2015). The observation error for in situ observations is dominated by the forward model error while that for GOSAT is dominated by the measurement error.

We do not have sufficient objective information to quantify the error correlation structure of $S_O$ and we

therefore assume it to be diagonal. This may underestimate $S_O$ because of correlated transport and source aggregation errors in the forward model, as noted above. We follow Zhang et al. (2018) to introduce a regularization factor $\gamma$ for the observation terms in the cost function $J(x)$ (Eq. 1) to avoid either overfits or underfits that would result from underestimates of $S_O$ and $S_A$, respectively. Lu et al. (2021) showed that the optimal value of this regularization factor can be selected such that the sum of the *n* prior terms

in the posterior estimate of the cost function ($J_A(\hat{x}) = (\hat{x} - x_A)^T S_A^{-1} (\hat{x} - x_A)$) has a value $\approx n$, which is the expected value from the Chi-square distribution with *n* degrees of freedom. Here we find that $\gamma = 1$ is best for both the in-situ and GOSAT inversions (i.e., no weighting in the inversion). We also conduct a sensitivity inversion using $\gamma = 0.5$ for the GOSAT observation terms as adopted in Maasakkers et al. (2021).


Table 2 summarizes the settings of our base inversion (in bold) and the inversion ensemble. The ensemble comprises 33 inversions using the different combinations of settings in the Table. The base inversion including GOSAT + in situ data represents our best estimate, but we will compare it prominently to the GOSAT-only and in-situ–only inversions with the same inversion parameters in order to evaluate the





contributions from the different observing platforms for optimizing emissions. We will use the other
ensemble members to discuss the sensitivity of inversion results to the choices of observations and
inversion parameters, and to define the range of uncertainty in the inversion results.

**2.6 Sectoral attribution and aggregation of inversion results**

The inversion returns the posterior estimates of mean emissions for each of the Gaussians, and we allocate
these emissions to the native $0.5^o \times 0.625^o$ model grid by summing the contributions of all Gaussians on
the grid. This defines a correction factor $f_0$ to total prior emissions for each $0.5^o \times 0.625^o$ grid cell and
including the contributions from all $q$ emission sectors (in our case $q = 12$, cf. Table 1). For sectoral
attribution of this total correction factor we need to derive the correction factors $f_i$ to the individual sectors
$i \in [1, q]$ contributing to $f_0$. We use two alternative methods for this purpose. The first method simply
takes $f_i = f_0$ for all $i$, thus assuming that all sectors contribute equally to the grid-level correction factor
(Maasakkers et al., 2021; Lu et al., 2021; Zhang et al., 2021) . The second method (Shen et al., 2021)
accounts for emissions from different sectors having different prior error standard deviation $\sigma_i$ and
therefore contributing differently to $f_0$. Following Shen et al. (2021), $f_i$ is then given by:

$$f_i = 1 - \frac{\eta \alpha_i \sigma_i^2 (1 - f_0)}{\sigma_A^2}, \quad (8)$$

where $\alpha_i$ is the fraction of total emissions in the grid cell contributed by sector $i$, $\sigma_A$ is the prior error

standard deviation for total emissions in the grid cell, and $\eta = \frac{\sigma_A^2}{\sum_{i=1}^{q} \alpha_i^2 \sigma_i^2}$ is a normalization factor. For the

prior error standard deviations $\sigma_i$ on the $0.5^o \times 0.625^o$ grid we use the scale-dependent adaptation by
Maasakkers et al. (2016) of EPA sectoral national error estimates. This results in prior error standard
deviations of 43% for rice, 66% for wastewater, 51% for landfills, 38% for livestock, 18% for coal, 30%
for gas, and 87% for oil emissions. We further use 70% for wetlands (Bloom et al., 2017) and 100% for
all other natural sources. These error estimates are solely used to infer $f_i$ values in equation (8), so that
more uncertain emissions will contribute more to the correction. We use the second method in our base
attribution of posterior estimates to emission sectors but will also use the results from the first method to
contribute to error ranges in these sector-attributed posterior estimates.

We also need to aggregate posterior emission estimates nationally and by sector for comparison to the
national emission inventories. Following Maasakkers et al. (2019), this is done by a transformation from
the posterior full-dimension state vector $\hat{x}$ to the reduced state vector $\hat{x}_{red}$ (national emission for a
given sector) with a summation matrix $W$:

$$\hat{x}_{red} = W\hat{x} \quad (9).$$

The posterior error covariance and averaging kernel matrices for the reduced state vector are then given
by

$$\hat{S}_{red} = W\hat{S}W^T \quad (10),$$
$$A_{red} = WAW^* \quad (11),$$

where $W^* = W^T(W \, W^T)^{-1}$ (Calisesi et al., 2005). $\hat{S}_{red}$ enables us to determine whether national
correction factors for individual sectors are affected by error correlations between sectors. $A_{red}$ enables
us to determine the ability of the observing system to quantify national emissions from a particular sector





independently from the prior estimate.


### 3. Results and Discussion

### 3.1 Base inversion compared to GOSAT-only and in-situ-only inversions

Figure 3a shows the gridded posterior correction factors from the base inversion averaged over 2010-2017, i.e., the multiplicative factors applied to the total prior emissions from Figure 2, mapped on the

$0.5°×0.625°$ model grid. Figure 3b shows the corresponding averaging kernel sensitivities, indicating the dependence of the posterior solution on the prior estimate (0 = total dependence, 1 = no dependence). The DOFS = 114 can be placed in the context of the 600 Gaussian state vector elements used to optimize the spatial distribution of emissions. We see that the observations provide considerable information to optimize methane emissions but we also see that a finer resolution for the inversion would not be justified

on the continental scale.

Figures 3c-f show the results from the GOSAT-only and in-situ-only inversions, enabling us to compare the information contents and consistency of the two data sets. The GOSAT-only inversion yields a DOFS of 68, while the in-situ-only inversion yields a DOFS of 80, even though there are 50% fewer in situ

observations than GOSAT observations. This is because the sensitivities of surface observations to emissions are an order of magnitude higher than those of satellite observations (Cusworth et al., 2018). The GOSAT observations have the advantage of broader coverage. Thus we find that the in-situ observations dominate the information content of the base inversion over California, the upper Midwest, and Canada; whereas GOSAT dominates the information content in Mexico (where there are no in-situ

observations) and most of the western US. GOSAT and in-situ observations contribute comparably in the south-central and eastern US, though with different weights in different locations. We conclude that GOSAT and in situ observations make comparable and complementary contributions to the optimization of methane emissions for North America.

We next examine the consistency in the information from GOSAT and in situ observations for correcting prior methane emissions. Inspection of the posterior correction factors from the GOSAT-only and in-situ-only inversions in Figure 3 shows overall qualitative agreement. Figure 4 displays a more quantitative comparison of the posterior corrections by correlating the values for $0.5°×0.625°$ grid cells between the GOSAT-only and in-situ-only inversions, selecting regions with relatively high averaging kernel

sensitivities for both. We find overall good consistency between the two inversions (correlation coefficient $r = 0.47$ for the ensemble of points, with 73% of grid cells showing corrections in the same direction). The reduced-major-axis regression slope is 0.62, consistent with GOSAT providing overall less information. Both inversions find that methane emissions over the south-central US, the southeast US, the Great Plains, and Alberta are underestimated in the prior inventories. They also agree on downward

corrections over central Canada and the Upper Midwest where wetland emissions dominate. The largest inconsistency is over California where the two inversions show correction factors in opposite direction for much of the state. This may reflect the underestimation of $CO_2$ over the Los Angeles Basin in the proxy GOSAT retrieval (Turner et al., 2015) and/or complex topography. Results from the base inversion tend toward either of the two inversions depending on which has the most information content.






We evaluated the ability of the base GOSAT + in situ inversion to fit the two observational data sets by comparing 2010-2017 GEOS-Chem simulations with posterior versus prior emissions and boundary conditions. Results are shown in Figure 5. The posterior simulation reduces the model mean bias (MB) at surface and tower measurements from -11 ppb in the prior simulation to -5 ppb, and also narrows the root-

mean-square error (RMSE) from 24 to 14 ppb. For GOSAT the improvement is less apparent from the comparison statistics, because the prior simulation already has a low mean bias MB = -0.5 ppb, and the prior RMSE is only 6.9 ppb (which decreases to 6.5 ppb). However, we see from Figure 5 a significant whitening of the noise with reduction of regional-scale biases.

**3.2 Optimized 2010-2017 anthropogenic methane emissions for CONUS, Canada, and Mexico**

Tables 1a-c summarize our inversion results for national 2010-2017 methane emissions by sector in CONUS, Canada, and Mexico. Our best estimates of total anthropogenic + natural emissions from the base inversion are 46.3 (40.2-48.4) Tg a$^{-1}$ for CONUS, 16.2 (13.5-17.4) Tg a$^{-1}$ for Canada, and 6.8 (5.4-6.9) Tg a$^{-1}$ for Mexico. The ranges given in parentheses are from the 33 inversion ensemble members

(Table 2). Averaging kernel sensitivities for these total national emissions (the diagonal elements in $\boldsymbol{A_{red}}$, section 2.6) are 0.72 for CONUS, 0.60 for Canada, and 0.40 for Mexico, indicate that the GOSAT + in situ observation system informs 72% of the total methane emissions in CONUS, 60% in Canada, and 40% in Mexico. The lower values for Mexico are due to the lack of in situ observations.

We partition these national totals into different sectors as described in Section 2.6, and use the posterior error covariance matrix (equation (10)) to evaluate the ability of the inversion to separate between sectors. This is shown in Figure 6 as the posterior error correlation matrix, displaying the error correlation coefficients ($r$) in the inversion results for all sector pairs. Error correlation coefficients are generally lower than 0.2 for CONUS, indicating successful separation, except for small sources (termites, seeps,

other anthropogenic). The same holds for Canada except for error correlation between landfills and wastewater treatment, both associated with urban areas. Anthropogenic emissions in Canada are well separated from the large wetland emissions. Error correlations are higher in Mexico, because emissions from different sectors tend to be concentrated in Mexico City and the eastern part of the country (Scarpelli et al., 2020), but even there the error correlation coefficients are generally less than 0.4. Optimization of

the oil/gas sector is well separated from the other sectors in all three countries.

We find that anthropogenic methane emissions for all three countries are larger in our inversion results than in the national inventories submitted to the UNFCCC. Our best estimate of the mean 2010-2017 anthropogenic methane emission for CONUS is 36.9 (32.5-37.8) Tg a$^{-1}$, which is 30% higher than the

28.7 Tg a$^{-1}$ in the 2016 version of the EPA GHGI used as prior estimate (EPA, 2016), and 42% higher than the mean 26.0 Tg a$^{-1}$ for 2010-2017 in the most recent version of the GHGI (EPA, 2021). Maasakkers et al. (2021) previously obtained a mean 2010-2015 CONUS anthropogenic emission of 30.6 (29.4–31.3) Tg a$^{-1}$ by inversion of GOSAT data using the same prior anthropogenic estimate as ours but a much higher prior estimate for CONUS wetlands (15.7 Tg a$^{-1}$). The need to decrease the wetlands source in their

inversion (to a posterior estimate of 11.8 Tg a$^{-1}$), as well as their reliance of GOSAT observations only,



may have dampened their ability to quantify anthropogenic emissions.

Our best estimate of the mean 2010-2017 anthropogenic methane emission for Canada is 5.3 (3.6-5.7) Tg a$^{-1}$, which is 43% higher than the 3.7 Tg a$^{-1}$ in the ECCC NIR (2020 version) used as prior estimate, and

33% higher than the 4.0 Tg a$^{-1}$ for 2010-2017 reported in the most recent version of the ECCC NIR (ECCC, 2021). Baray et al. (2021) previously obtained a mean 2010-2015 anthropogenic emission of 6.1 Tg a$^{-1}$ for Canada by inversion of data from GOSAT and ECCC surface sites.

Our best estimate of the mean 2010-2017 anthropogenic methane emission for Mexico is 6.0 (4.7-6.1) Tg

a$^{-1}$, which is 20% higher than the 5.0 Tg a$^{-1}$ in Mexico's national inventory (INECC and SEMARNAT, 2018) used as prior estimate. Shen et al. (2021) similarly found higher emissions than the national inventory in their inversion of TROPOMI methane data for eastern Mexico.

Figure 7 displays the data from Tables 1a-c for the national posterior emission estimates from different

sectors in comparison with the EPA (US), ECCC (Canada), and INECC (Mexico) national inventories used as prior estimates. We find that emissions from all major sectors except coal and wastewater are lower in the national inventories than our inversion results, with the largest underestimates for fugitive emissions from the oil sector. The total CONUS oil and gas emissions in our inversion are 4.6 and 9.9 Tg a$^{-1}$, respectively, 109% and 45% higher than the EPA (2016) inventory used here as prior estimate, and

177% and 65% higher than the most recent EPA (EPA, 2021) inventory for the 2010-2017 mean. The EPA inventory reports an uncertainty of -24 to +29% for oil and -15 to +14% for natural gas emissions (EPA, 2021). Our estimates are also higher than those in Maasakkers et al. (2021), which are 3.6 and 8.0 Tg a$^{-1}$ respectively for oil and gas emissions in 2010-2015. They are consistent with the Alvarez et al. (2018) estimates for total CONUS oil and gas emissions of 13 (11–15) Tg a$^{-1}$ in 2015 based on field

measurements within oil and gas basins, scaled up to derive a national value.

We mentioned previously that the lower estimates in Maasakkers et al. (2021) could reflect their use of GOSAT observations only, the difference in time frame, and their high prior estimate for wetlands, but another factor is their assumption of normal distributions for prior emission error standard deviations. We

find from our inversion ensemble that assuming a log normal distribution instead (as in our base inversion) increases the resulting posterior oil and gas emissions by 0.8 and 0.9 Tg a$^{-1}$ respectively by better capturing the heavy tail of the emission probability density functions, as previously observed in oil/gas production regions (Zavala-Araiza et al., 2015; Frankenberg et al., 2016; Alvarez et al., 2018). Adding the in situ observations to the GOSAT-only inversion further increases the posterior oil and gas emissions

by 0.2 and 0.3 Tg a$^{-1}$, respectively. Thus our base inversion yields the high end of the estimated range from the inversion ensemble (Table 1a) but still represents our best estimate.

Our inversion increases the oil emissions over Canada by more than a factor of two to 1.8 Tg a$^{-1}$ compared to the ECCC inventory. The total posterior oil and gas emissions for Canada are 2.9 (1.6-3.3) Tg a$^{-1}$. This

is in good agreement with a recent inversion study (3.0 Tg a$^{-1}$) based on 2010-2017 surface methane measurements in western Canada (Chan et al., 2020). Most of the information for Canada in our base



inversion indeed comes from the in situ measurements (Figure 3), which are relatively dense in Canada (Figure 1), and considering that GOSAT observations at high latitudes are relatively sparse and seasonally limited (Lu et al., 2021). Maasakkers et al. (2021) previously found little information for Canadian anthropogenic emissions in their GOSAT-only inversion, although that was further complicated by their large overestimate of prior wetland emissions that dominate total emissions in Canada.

We further compared our oil/gas inversion results for CONUS, Canada, and Mexico to the TRACE bottom-up inventory aggregating data from individual assets up to the country level (Climate TRACE, 2021). This inventory uses lifecycle assessment emissions models for production, processing, refining, and shipping (Gordon et al., 2015; Masnadi et al., 2018; Gordon and Reuland, 2021). The TRACE oil/gas total emission estimates for CONUS (9.6 Tg a$^{-1}$), Canada (1.8 Tg a$^{-1}$), and Mexico (0.5 Tg a$^{-1}$) are similar to the prior estimates from EPA, ECCC, and INECC respectively (Table 1) and correspondingly lower than our best posterior estimates of 14.5 Tg a$^{-1}$ for CONUS, 3.2 Tg a$^{-1}$ for Canada, and 1.3 Tg a$^{-1}$ for Mexico. The bottom-up oil and gas modeling in TRACE assesses routine methane emissions from normal operations, assuming normal fugitive emissions. Recent flyover work, however, shows that methane emissions are highly intermittent (Cusworth, et. al., 2021) and this is not well captured in bottom-up estimates.

Figure 8 shows the spatial distributions of posterior correction to the gridded version of national inventories for the oil, gas, livestock, and landfill sectors. We find large upward corrections for the major oil/gas production basins in the US including the Permian, Barnett Shale, Eagle Ford, Bakken Shale, Marcellus Shale, and Anadarko basins, consistent with previous reports based on field measurements and satellite observations (Miller et al., 2013; Karion et al., 2015; Peischl et al., 2015; Lyon et al., 2015; Ren et al., 2019; Robertson et al., 2020; Zhang et al., 2020). Upward corrections in Canada are concentrated over the oil/gas production regions of Alberta and Saskatchewan, again consistent with previous studies (Johnson et al., 2017; Baray et al., 2018; Chan et al., 2020). For Mexico the upward correction is concentrated in the onshore Sureste Basin which is the largest oil field in the country, but with a downward correction for offshore operations. This is consistent with aircraft and TROPOMI satellite observations, suggesting that methane from offshore oil platforms is piped onshore and inefficiently flared (Zavala-Araiza et al., 2021; Shen et al., 2021).

The spatial distribution of posterior corrections to livestock emissions indicates that the national inventories are too low for most regions, although there are exceptions in particular in the western US (Fig. 8). The highest emissions in the gridded version of the EPA (2016) GHGI are for the Upper Midwest and our inversion results suggest that these are too low, possibly reflecting higher-emitting manure management systems from confined animal feeding operations than included in the GHGI calculations (Sheng et al., 2018a). Yu et al. (2021) also found from an aircraft-based inversion that livestock emissions from the EPA inventory over the US Corn Belt and Upper Midwest region are underestimated by 25% during summer and winter.

Our inversion finds CONUS methane emissions from landfills of 7.2 (6.0-7.6) Tg a$^{-1}$, 24% higher than





the prior EPA (2016) estimate of 5.8 Tg a$^{-1}$. The most recent EPA (2021) inventory gives 4.5 Tg a$^{-1}$ for landfill emissions with an uncertainty of ±22%. The organic decay rate and methane production potential used in the GHGI calculation may be too low (Wang et al., 2013; Sun et al., 2019).

### 3.3 2010-2017 trends in anthropogenic methane emissions

Our inversion optimizes emissions for individual years in 2010-2017, allowing investigation of emission trends. Figure 9 shows the 2010-2017 time series of total anthropogenic methane emissions from CONUS, Canada, and Mexico, and the contributions from the dominant sectors (oil, gas, coal, livestock, and landfills). We include no trend in the prior estimates so that the trends in Figure 9 are solely driven by the observations. Table 3 gives the corresponding 2010-2017 linear trends in emissions inferred from ordinary linear regression, and compares to the trends reported in the most recent national inventories for the US (EPA, 2021) and Canada (ECCC, 2021). Mexico only reports emissions up to 2015.

Our inversion shows that over the time frame of 2010 to 2017, total anthropogenic methane emissions in CONUS peaked in 2014 and then decreased, resulting in no net trend for the 2010-2017 period (0.1 (-0.1-0.3) % a$^{-1}$). The increasing trend for 2010-2015 is 0.9 (0.4-1.8) % a$^{-1}$, higher than 0.4 % a$^{-1}$ in the GOSAT-only inversion by Maasakkers et al. (2021) and more consistent with the 0.7±0.3 % a$^{-1}$ for 2006-2015 in an in-situ-only inversion by Lan et al. (2019). Inspection of CONUS trends for different emission sectors in the base inversion indicates that the 2014 maximum largely reflects opposite trends between oil and landfill emissions, which increased by 2.9 (1.0-2.9) % a$^{-1}$ and 1.7 (1.0-1.8) % a$^{-1}$ respectively over the 2010-2017 period, and gas emissions, which decreased by 1.8 % a$^{-1}$ over the 2010-2017 period, with livestock and coal emissions showing no significant trend (Figure 9). In contrast, the most recent EPA GHGI inventory reports a steady decreasing trend of -0.8 % a$^{-1}$ in US anthropogenic methane emissions over the 2010-2017 period mostly driven by coal (-5.4% a$^{-1}$) and landfills (-1.6% a$^{-1}$) (EPA, 2021). The decrease for gas is more pronounced in our inversion than in the EPA inventory (-0.4% a$^{-1}$). The EPA inventory shows no significant trend for oil emissions.

Figure 10 shows the spatial distributions of the linear regression fits to the 2010-2017 trends for the major anthropogenic sectors, i.e., the equivalent linear trends over the period. We find that the oil increases are mostly driven by major basins in the south-central US including the Permian and Eagle Ford basins. The gas decreases are mostly driven by fields in the western US (Niobrara) and southeastern US (Haynesville). Livestock emissions show variable regional patterns of increase and decrease that could reflect variations in animal populations. The increase in Iowa is consistent with a previous study of GOSAT trends by Sheng et al. (2018a), who attributed it to an increase in the number of swine (Iowa Department of Natural Resources, 2017). Landfills also show variable patterns of increase and decrease.

The ECCC reports no significant trends of Canadian anthropogenic methane emissions over 2010-2017, but notes some decreases of oil emissions, in particular after 2014 (ECCC, 2021). Here we find a decreasing trend in Canadian anthropogenic emissions of -2.3 (-2.5 - -1.6) % a$^{-1}$ from the inversion, mainly driven by gas (-3.8 (-3.9 - -1.7) % a$^{-1}$) and oil (-1.7 (-2.0 - -0.4) % a$^{-1}$). This may reflect reductions in livestock and oil/gas emissions over this period (ECCC, 2020b) and the ongoing regulation of methane





released from the oil/gas sectors following the Pan-Canadian Framework on Clean Growth and Climate
Change, which aims to reduce methane emissions by 40-45% by 2025 relative to the 2012 level (ECCC,
2017). The inversion also suggests a decreasing trend in Mexican anthropogenic methane emissions by -
3.3 (-3.4 - -1.7) % a$^{-1}$, but this is mainly driven by a decrease from 2010 to 2011. We find very large
relative decreases of oil emissions (-11.6 (-15.0 - -3.5) % a$^{-1}$) in particular for offshore Sureste, consistent
with increasing utilization of associated gas (Zhang et al., 2019).


### 3.4 2010-2017 wetland methane emissions and trends

Our inversion shows strong ability to optimize wetland emissions over CONUS and Canada (averaging
kernel sensitivity of 0.57). Wetland emissions in Mexico are much smaller and are not efficiently
optimized by the inversion, as shown in Table 1c. Posterior wetland emissions are 8.4 (6.4-10.6) Tg a$^{-1}$
for CONUS and 9.9 (7.8-12.0) Tg a$^{-1}$ for Canada, compared to 7.5 and 12.0 Tg a$^{-1}$ in the prior estimate
from the WetCHARTs v1.3.1 high-performance subset for North America (Ma et al., accepted). There are
larger regional upward (southeast US) and downward (Upper Midwest) corrections even with this high-
performance subset, as shown in Figure 11, pointing to major gaps in our understanding.

Figures 11 and 12 show the 2010-2017 trends of wetland emissions for CONUS and Canada. We find a
significant increase of 2.6 (1.7-3.8) % a$^{-1}$ in wetland methane emissions over CONUS in 2010-2017, in
particular after 2014, and this is consistent with but higher than the WetCHARTs trend estimates of 1.3 %
a$^{-1}$ (not used in the inversion). The trends over CONUS are mostly driven by increases in the southeast
US (Fig.11b). Fluctuations in emissions for temperate and boreal wetlands are mostly modulated by
temperature, snow melt, precipitation, and drought events (Watts et al., 2014). We find a significant
correlation of 0.89 between the CONUS wetland emissions and annual precipitation in the CONUS
wetland regions, and a strong 2010-2017 increase in the later that may drive the wetland trends (Fig.12c).
Wetland emissions over Canada do not show significant trends in the inversion. The 2016 peak is
consistent with WetCHARTs and may be explained by high precipitation (Fig.12d).


### 4 Conclusions

We estimated mean methane emissions and trends for 2010-2017 in the contiguous United States
(CONUS), Canada, and Mexico by inversion of in situ (GLOBALVIEWplus CH$_4$ ObsPack) and satellite
(GOSAT) atmospheric methane observations. Our inversion used gridded versions of the national
anthropogenic emission inventories reported to the UNFCCC by EPA (CONUS), ECCC (Canada) and
INECC (Mexico) as prior estimates. It optimized a 600-member Gaussian mixture model (GMM) of
emissions for individual years at up to 0.5×0.625° resolution. The inversion involved analytic
minimization of the Bayesian cost function with log-normal prior statistics. This enabled a large ensemble
of inversions to test the sensitivity of results to a range of assumptions, and provided closed-form
expressions of posterior error covariance and information content to evaluate the results for different
regions and emission sectors. We find that GOSAT and in situ observations make comparable and
complementary contributions to the optimization of methane emissions for North America, and that they
show overall consistent corrections to prior methane emissions.



We estimate from our base inversion a mean 2010-2017 methane emission of 46.3 (40.2-48.4 ensemble
       range) Tg a$^{-1}$ for CONUS, of which 36.9 (32.5-37.8) Tg a$^{-1}$ is anthropogenic. This anthropogenic emission
       is 30% higher than the EPA inventory of 28.7 Tg a$^{-1}$ used as prior estimate (EPA, 2016), and 42% higher
       than the 2010-2017 mean of 26.0 Tg a$^{-1}$ in the most recent version of the EPA inventory (EPA, 2021).
       These upward corrections are largely attributed to the oil (4.6 Tg a$^{-1}$) and gas (9.9 Tg a$^{-1}$) sectors, which
are respectively 177% and 65% higher than the EPA (2021) estimates. The upward corrections of the oil
       and gas sectors are mainly in large basins of the south-central US. The inversion also shows upward
       corrections of livestock emissions to 10.6 Tg a$^{-1}$, 15% higher than the EPA estimate (9.2 Tg a$^{-1}$), and of
       landfill emissions to 7.2 Tg a$^{-1}$, 24% higher than the EPA estimate (5.8 Tg a$^{-1}$).

We estimate a mean 2010-2017 emission for Canada of 16.2 (13.5-17.4) Tg a$^{-1}$, of which 5.3 (3.6-5.7) Tg
       a$^{-1}$ is anthropogenic. This anthropogenic emission is 43% higher than the 3.7 Tg a$^{-1}$ in the ECCC (2020)
       national inventory used as prior estimate. Most of this difference is due to oil emissions which we estimate
       at 1.8 Tg a$^{-1}$, more than twice the ECCC estimate, and mainly from production in Alberta and
       Saskatchewan.


       We estimate a mean 2010-2017 emission for Mexico of 6.8 (5.4-6.9) Tg a$^{-1}$, of which 6.0 (4.7-6.1) Tg a$^{-1}$
       is anthropogenic. This anthropogenic emission is 20% higher than the 5.0 Tg a$^{-1}$ in the INECC (2018)
       national inventory used as prior estimate. Again, most of the underestimate is due the oil sector and
       specifically to oil production in the Sureste onshore region. Offshore oil emissions are lower than the
INECC estimate, suggesting that the associated gas is piped onshore and then vented, perhaps because of
       inefficient flaring.

       We find from the inversion that anthropogenic emissions in CONUS peaked in 2014 and had no net trend
       over the 2010-2017 period (0.1 (-0.1-0.3) % a$^{-1}$), in contrast with the EPA inventory that reports a steady
decreasing trend of -0.8 % a$^{-1}$ over this period. The net trend in the inversion reflects compensating effects
       from increases in emissions from the oil and landfill sectors, decreases from the gas sector, and flat
       emissions from the livestock and coal sectors. We find a decreasing trend in Canadian anthropogenic
       emissions of -2.3 (-2.5 - -1.6) % a$^{-1}$ over the 2010-2017 period, mainly driven by oil and gas production.
       We also find a decreasing trend in Mexican anthropogenic methane emissions (-3.3 (-3.4 - -1.7) % a$^{-1}$)
over the 2010-2017 period, mostly driven by the oil sector and in particular by offshore operations.

       Wetlands are the main natural source of methane in all three countries. Starting from the high-performance
       subset of the WetCHARTs inventory ensemble as prior estimate, our inversion yields mean wetland
       emission estimates for 2010-2017 of 8.4 (6.4-10.6) Tg a$^{-1}$ for CONUS, 9.9 (7.8-12.0) Tg a$^{-1}$ for Canada,
and 0.6 (0.4-0.6) Tg a$^{-1}$ for Mexico. Wetland emissions in CONUS show a significant increase of 2.6 (1.7-
       3.8) % a$^{-1}$ over 2010-2017 correlated with precipitation.

       **Data availability.** The GLOBALVIEWplus CH4 ObsPack v1.0 data product is available at
       https://gml.noaa.gov/ccgg/obspack/data.php?id=obspack_ch4_1_GLOBALVIEWplus_v1.0_2019-01-
08. The University of Leicester GOSAT Proxy v9.0 XCH4 data is available from the Centre for



Environmental Data Analysis data repository at
http://dx.doi.org/10.5285/18ef8247f52a4cb6a14013f8235cc1eb (Parker and Boesch, 2020). The
WetCHARTs v1.3.1 is available at the ORNL DAAC, https://doi.org/10.3334/ORNLDAAC/1915.
Modeling data can be accessed by contacting the corresponding author Xiao Lu
(luxiao25@mail.sysu.edu.cn).

**Author contributions.** XL and DJJ designed the study. XL conducted the modeling and data analyses
with contributions from HLW, JDM, ZYZ, TRS, LS, ZQ, MPS, and HN. AAB, SM, and JRW contributed
to theWetCHARTs wetland emission inventory and its interpretation. RJP and HB contributed to the
GOSAT satellite methane retrievals. All authors provided insightful comments. XL and DJJ wrote the
paper with input from all authors.

**Competing interests.** The authors declare that they have no conflict of interest.

**Acknowledgments.** We    acknowledge    all    of    the    data    providers    and    laboratories
(https://search.datacite.org/works/10.25925/20190108, last access: 7 July 2021) that contributed to the
GLOBALVIEWplus CH$_4$ ObsPack v1.0 data product compiled by NOAA Global Monitoring Laboratory.
We thank the Japanese Aerospace Exploration Agency, National Institute for Environmental Studies, and
the Ministry of Environment for the GOSAT data and their continuous support as part of the Joint
Research Agreement. This research used the ALICE High Performance Computing Facility at the
University of Leicester for the GOSAT retrievals. Part of the research was carried out at the Jet Propulsion
Laboratory, California Institute of Technology, under a contract with the National Aeronautics and Space
Administration. We thank Arlyn Andrews from the National Oceanic and Atmospheric Administration for
contribution to the GLOBALVIEWplus CH4 ObsPack v1.0 data product. We thank Bill Irving, Erin
McDuffie, and Melissa Weitz from the United States Environmental Protection Agency for helpful
comments.

**Financial support.** This work was funded by the NOAA Atmospheric Chemistry, Carbon Cycle and
Climate (AC4) Program and by the NASA Carbon Monitoring System. RJP and HB are funded via the
UK National Centre for Earth Observation (NCEO grant numbers: NE/R016518/1 and NE/N018079/1).
We acknowledge funding from the ESA GHG-CCI and Copernicus C3S projects.





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





**Table 1a.** Mean 2010-2017 methane emissions for the contiguous US (CONUS)

| | Prior [a] | Posterior [b] | Sensitivity [c] |
|---|---|---|---|
| **Total sources [Tg a$^{-1}$]** | 36.8 | 46.3 (40.2-48.4) | 0.72 |
| **Anthropogenic sources** | 28.7[d] | 36.9 (32.5-37.8) | 0.55 |
| Livestock | 9.2 | 10.6 (9.2-11.8) | 0.43 |
| Oil | 2.3 | 4.6 (3.0-4.7) | 0.43 |
| Natural gas | 6.8 | 9.9 (8.1-10.5) | 0.42 |
| Coal mining | 2.9 | 2.8 (2.4-3.5) | 0.44 |
| Landfills | 5.8 | 7.2 (6.0-7.6) | 0.34 |
| Wastewater | 0.70 | 0.63 (0.56-0.74) | 0.57 |
| Rice cultivation | 0.48 | 0.65 (0.49-0.68) | 0.33 |
| Other Anthropogenic | 0.46 | 0.45 (0.44-0.54) | 0.30 |
| **Natural Sources** | 8.6 | 9.5 (7.4-11.5) | 0.64 |
| Wetlands | 7.5 | 8.4 (6.4-10.6) | 0.57 |
| Open fires | 0.16 | 0.17 (0.15-0.24) | 0.43 |
| Termites | 0.59 | 0.63 (0.57-0.76) | <0.1 |
| Seeps | 0.28 | 0.27 (0.23-0.35) | 0.14 |

[a] Prior estimates for the inversion. Anthropogenic emissions are from the Environmental Protection Agency (EPA) Inventory of U.S. Greenhouse Gas Emissions and Sinks (GHGI) for year 2012 as reported by EPA (2016). Wetland emissions are the 2010-2017 mean of the high-performance subset of the WetCHARTs ensemble (Ma et al., accepted). Open fire emissions area from GFEDv4s (van der Werf et al., 2017). Termite and seep emissions are as described in Lu et al. (2021).

[b] Results from the base inversion of GOSAT and GLOBALVIEWplus data, with the range from the inversion ensemble and from the two sectoral attribution methods (66 total ensemble members) in parentheses.

[c] Sensitivity of the posterior estimate to the observations as diagnosed by the diagonal elements of the averaging kernel matrix, ranging from 0 (no sensitivity, posterior equal to prior) to 1 (full sensitivity, posterior fully determined by the observations). Values are from the base inversion for year 2015. Results for other years show similar values. See section 2.6 for more details.

[d] The most recent EPA GHGI report (EPA, 2021) gives a mean anthropogenic emission of 26.0 Tg a$^{-1}$ for 2010-2017. Anthropogenic US emissions outside CONUS (mostly Alaska) account for only 0.3 Tg a$^{-1}$ according to the Maasakkers et al. (2016).





**Table 1b.** Mean 2010-2017 methane emissions for Canada.

|  | Prior [a] | Posterior [b] | Sensitivity [c] |
|---|---|---|---|
| **Total sources [Tg a$^{-1}$]** | 17.1 | 16.2 (13.5-17.4) | 0.60 |
| **Anthropogenic sources** | 3.7 | 5.3 (3.6-5.7) | 0.59 |
| Livestock | 1.1 | 1.4 (1.0-1.6) | 0.48 |
| Oil | 0.75 | 1.8 (0.81-1.9) | 0.48 |
| Natural gas | 0.80 | 1.1 (0.76-1.6) | 0.54 |
| Coal mining | <0.1 | <0.1 | 0.51 |
| Landfills | 0.66 | 0.69 (0.45-0.74) | 0.33 |
| Wastewater | <0.1 | <0.1 | 0.20 |
| Rice cultivation | 0 | 0 | / |
| Other Anthropogenic | 0.27 | 0.31 (0.26-0.36) | 0.18 |
| **Natural Sources** | 13.5 | 10.9 (8.7-13.2) | 0.54 |
| Wetlands | 12.0 | 9.9 (7.8-12.0) | 0.57 |
| Open fires | 1.1 | 0.67 (0.48-0.95) | 0.54 |
| Termites | 0.28 | 0.29 (0.24-0.30) | <0.1 |
| Seeps | <0.1 | <0.1 | <0.1 |

[a] Prior estimates for the inversion. Anthropogenic emissions are from the Environment and Climate Change Canada (ECCC) National Inventory Report (NIR) for year 2018 (ECCC, 2020). Wetland emissions are the 2010-2017 mean of the high-performance subset of the WetCHARTs ensemble (Ma et al., 2021). Open fire emissions area from GFEDv4s (van der Werf et al., 2017). Termite and seep emissions are as described in Lu et al. (2021).

[b] Results from the base inversion of GOSAT and GLOBALVIEWplus in situ data, with the range from the inversion ensemble and from the two sectoral attribution methods (66 total ensemble members) in parentheses.

[c] Sensitivity of the posterior estimate to the observations as diagnosed by the diagonal elements of the averaging kernel matrix, ranging from 0 (no sensitivity, posterior equal to prior) to 1 (full sensitivity, posterior fully determined by the observations). Values are from the base inversion for year 2015. Results for other years show similar values. See section 2.6 for more details.





**Table 1c.** Mean 2010-2017 methane emissions for Mexico.

| | Prior [a] | Posterior [b] | Sensitivity [c] |
|---|---|---|---|
| **Total sources [Tg a⁻¹]** | 5.8 | 6.8 (5.4-6.9) | 0.40 |
| **Anthropogenic sources** | 5.0 | 6.0 (4.7-6.1) | 0.41 |
| Livestock | 2.3 | 2.5 (2.0-2.6) | 0.24 |
| Oil | 0.44 | 0.84 (0.42-0.85) | 0.20 |
| Natural gas | 0.34 | 0.42 (0.36-0.53) | 0.44 |
| Coal mining | 0.28 | 0.26 (0.26-0.52) | 0.80 |
| Landfills | 0.77 | 1.0 (0.67-1.0) | 0.30 |
| Wastewater | 0.69 | 0.80 (0.65-0.86) | 0.14 |
| Rice cultivation | <0.1 | <0.1 | <0.1 |
| Other Anthropogenic | 0.13 | 0.14 (0.12-0.16) | 0.10 |
| **Natural Sources** | 0.79 | 0.83 (0.64-0.89) | 0.10 |
| Wetlands | 0.52 | 0.57 (0.43-0.60) | <0.1 |
| Open fires | 0.14 | 0.14 (0.10-0.16) | <0.1 |
| Termites | 0.13 | 0.12 (0.10-0.14) | <0.1 |
| Seeps | <0.1 | <0.1 | <0.1 |

[a] Prior estimates for the inversion. Anthropogenic emissions are from the National Inventory of Greenhouse Gases and Compounds constructed by the Instituto Nacional de Ecología y Cambio Climático (INECC). Wetland emissions are the 2010-2017 mean of the high-performance subset of the WetCHARTs ensemble (Ma et al., 2021). Open fire emissions area from GFEDv4s (van der Werf et al., 2017). Termite and seep emissions are as described in Lu et al. (2021).

[b] Results from the base inversion of GOSAT and GLOBALVIEWplus data, with the range from the inversion ensemble and from the two sectoral attribution methods (66 total ensemble members) in parentheses.

[c] Sensitivity of the posterior estimate to the observations as diagnosed by the diagonal elements of the averaging kernel matrix, ranging from 0 (no sensitivity, posterior equal to prior) to 1 (full sensitivity, posterior fully determined by the observations). Values are from the base inversion for year 2015. Results for other years show similar values. See section

2.6 for more details.





**Table 2.** Settings for generation of the 33-member inversion ensemble [a].

| Observations | Regularization parameter $\gamma$ | Prior error standard deviation | |
| --- | --- | --- | --- |
| | | Emissions | Boundary conditions |
| **GOSAT + in situ** | **1** | **50% (log normal)** | **10 ppb** |
| GOSAT + in situ (long-term)[c] | 0.5 (GOSAT) | quadrature sum (log normal)[d] | 5 ppb |
| GOSAT | | 50% (normal) | |
| In situ | | 95% (normal) | |
| In situ (long-term) [c] | | quadrature sum (normal) [d] | |

[a] Settings for the base inversion are in bold. The 33-member inversion ensemble uses the different combinations of settings to probe the effects of different choices in observations and in inversion parameters. The GOSAT + in situ inversion includes the following 7-member ensemble: (1) base inversion with $\gamma = 1$ for in situ and GOSAT observations , $\sigma_A = 50\%$ (log normal) for emissions, and $\sigma_A = 10$ ppb for boundary conditions; (2) the same as (1) except that $\gamma = 0.5$ for GOSAT observations; (3)-(6) the same as (1), except that $\sigma_A$ for emissions uses the other 4 options in the Table; and (7) is the same as (1), except that $\sigma_A = 5$ ppb for boundary conditions. Similarly, the GOSAT + in situ (long-term) and GOSAT inversions have 7 ensemble members, respectively. The in situ and in situ (long-term) inversion have 6 ensemble members, respectively. This adds up to 33 inversion ensemble members. Sectoral attribution is done by two alternative methods (see text in Section 2.6), resulting in a total of 66 members.

[c] Including only long-term surface and tower sites with observations for all years of the 2010-2017 record.

[d] Adding the errors from individual sectors in quadrature following Maasakkers et al. (2021).





**Table 3.** 2010-2017 trends in methane anthropogenic emissions[a].

| | Inversion ensemble[b] | National inventories[c] |
|---|---|---|
| CONUS (% a$^{-1}$) | | |
| Total anthropogenic | 0.10 (-0.11 - 0.34) | -0.8* |
| Livestock | -0.25 (-0.61 - 0.09) | 0.5 |
| Oil | 2.9* (1.0 - 1.9) | -0.4 |
| Natural gas | -1.8* (-1.8 - -0.48) | -0.4* |
| Coal mining | -1.0 (-1.9 - -0.04) | -5.4* |
| Landfills | 1.7*(1.0-1.8) | -1.6* |
| Canada (% a$^{-1}$) | | |
| Total anthropogenic | -2.3 (-2.5 - -1.6) | -0.3 |
| Livestock | -2.2 (-2.7 - -1.5) | -0.3 |
| Oil | -1.7 (-2.0 - -0.42) | -1.2 |
| Natural gas | -3.8 (-3.9 - -1.7) | -0.1 |
| Landfills | -2.3 (-3.9 - -1.7) | -0.4* |
| Mexico (% a$^{-1}$) | | |
| Total anthropogenic | -3.3 (-3.4 - -1.7) | NA |
| Oil | -11.6* (-15.0 - -3.5) | NA |
| Natural gas | -3.1 (-6.1 - -1.0) | NA |

[a]From ordinary linear regression of emissions in individual years, reported in % a$^{-1}$ relative to the 2010-2017 mean.
Figure 9 shows the time series for the base inversion. Trends marked with * are significant with *p-value*<0.1.
[b]Trends from the base inversion, with the range of trends from the inversion ensemble members in parentheses.
[c]From national inventory emissions in individual years reported by the Environmental Protection Agency (EPA, 2021)
and Environment and Climate Change Canada (ECCC, 2021). INECC in Mexico only reports emissions up to 2015
hence the Not Available (NA) entries.





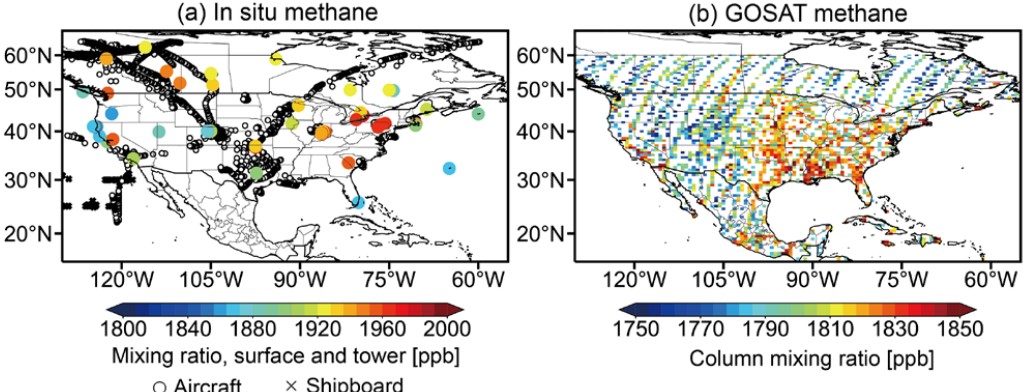

**Figure 1.** Methane observations over North America used in the inversion. The observations are from the in situ GLOBALVIEWplus $CH_4$ ObsPack data product and from the GOSAT satellite instrument. Mixing ratios shown for surface, tower, and GOSAT observations are means for 2010-2017. Aircraft and shipboard observation locations are shown as additional symbols. The GOSAT data are dry column mixing ratios from the University of Leicester version 9 Proxy $XCH_4$ retrieval (Parker et al., 2020a) and are averaged here on the $0.5° \times 0.625°$ GEOS-Chem model grid.




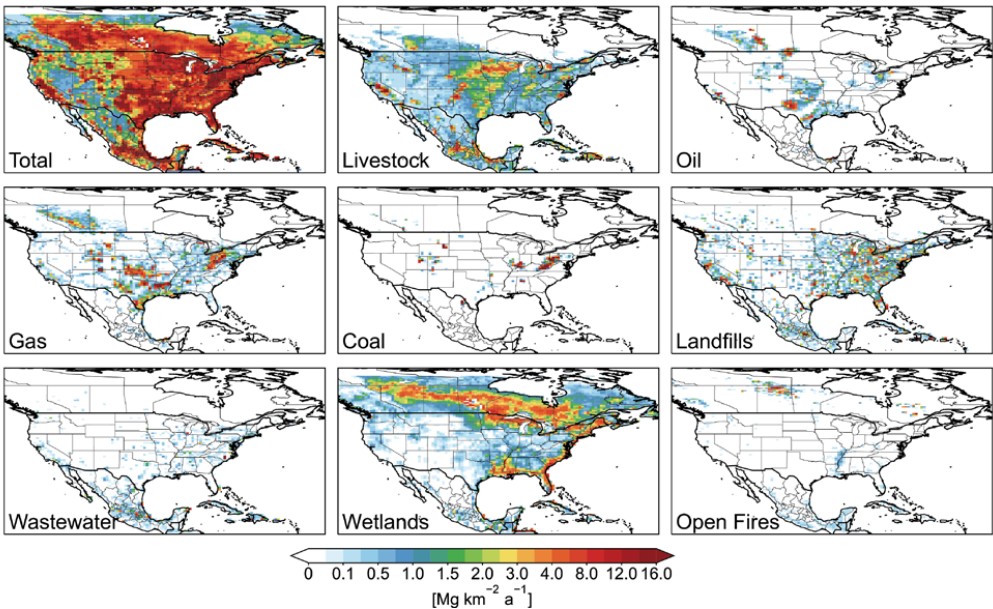

**Figure 2.** Prior estimates of methane emissions from individual sectors. Anthropogenic emissions are from spatially explicit versions of the EPA, ECCC, and INECC official national inventories. Wetland emissions are from the mean of the high-performance subset of the WetCHARTs inventory ensemble.


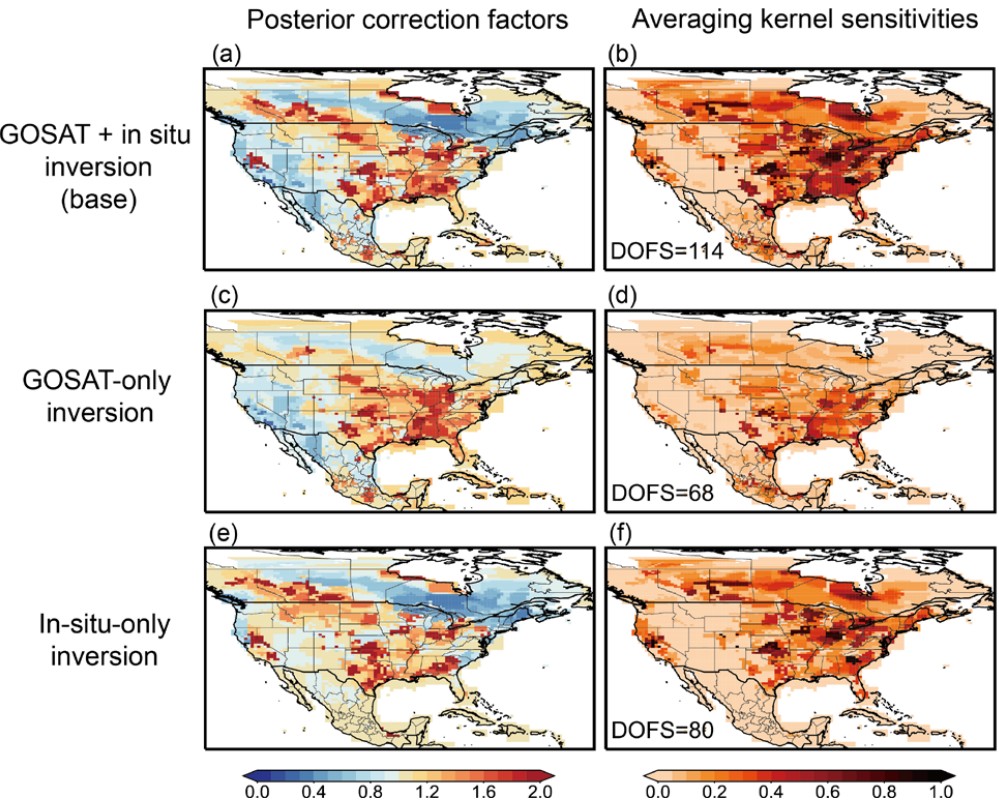

**Figure 3.** Optimization of mean 2010–2017 methane emissions over North America. Results are from the base inversion using both GOSAT and GLOBALVIEWplus in situ observations, the GOSAT-only inversion, and the in-situ only inversion. The left panels show the posterior correction factors, i.e., the multiplicative factors applied to the total prior emissions in Figure 2, and the right panels show the averaging kernel sensitivities (diagonal elements of the averaging kernel matrix). The degrees of freedom for signal (DOFS, defined as the trace of the averaging kernel matrix) are shown inset.

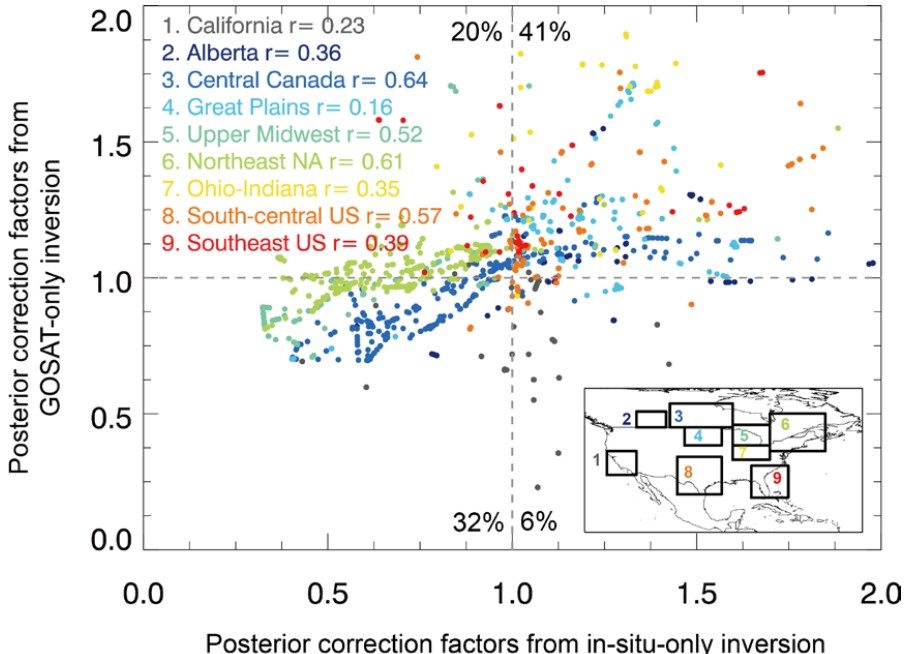

**Figure 4.** Comparison of posterior correction factors to prior methane emissions on the 0.5º × 0.625º grid between
GOSAT-only and in-situ-only inversions. The comparisons are for 9 regions with relatively high averaging kernel
sensitivities for both inversions. Each point represents the posterior correction factors from both inversions in a
0.5ºx0.625º grid cell (Fig.3). Correlation coefficients for each of the nine regions are shown inset. Percentiles in each
quadrant show the fraction of the total points in that quadrant.




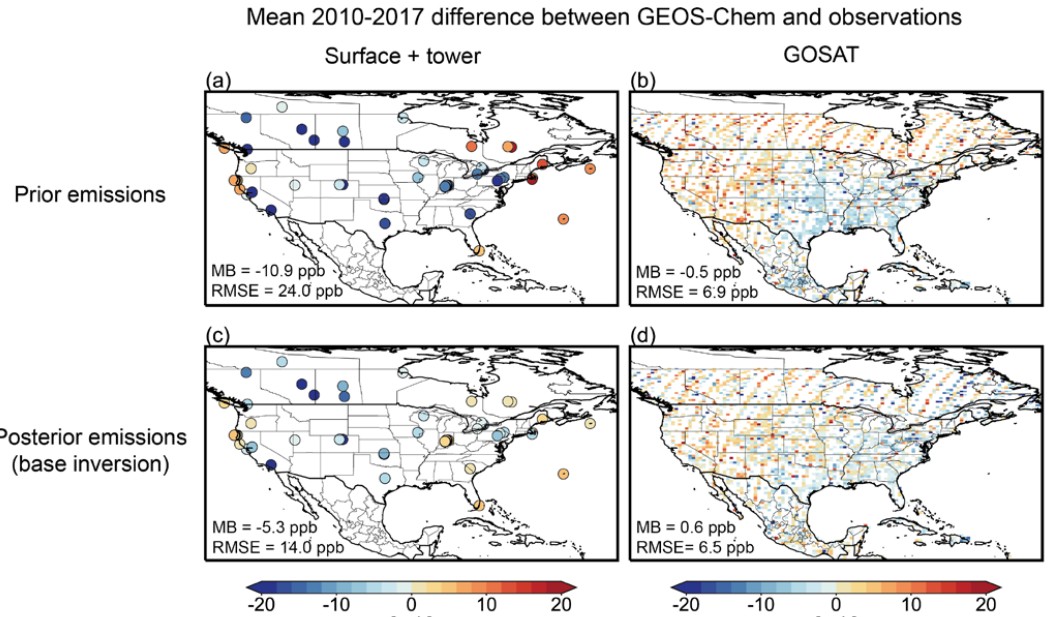

**Figure 5.** Ability of the base inversion to fit the in situ (surface and tower) and GOSAT observations for 2010-2017. The figure shows the mean differences between GEOS-Chem simulations and the observations using either prior or posterior methane emissions. Mean bias (MB) and root-mean square error (RMSE) are shown inset, calculated from the temporally

averaged differences for each in situ site or GOSAT grid cell.



**Figure 6.** Posterior error correlation coefficients (*r*) between sectoral methane emissions in the contiguous US (CONUS), Canada, and Mexico, using the sector-aggregated error covariance matrix as described in Section 2.6. Error correlation coefficients indicate the ability of the inversion to separate emissions between sectors (0 = perfectly, ± 1 = not at all). Results are from the base inversion for year 2015. Results for other years show similar patterns.

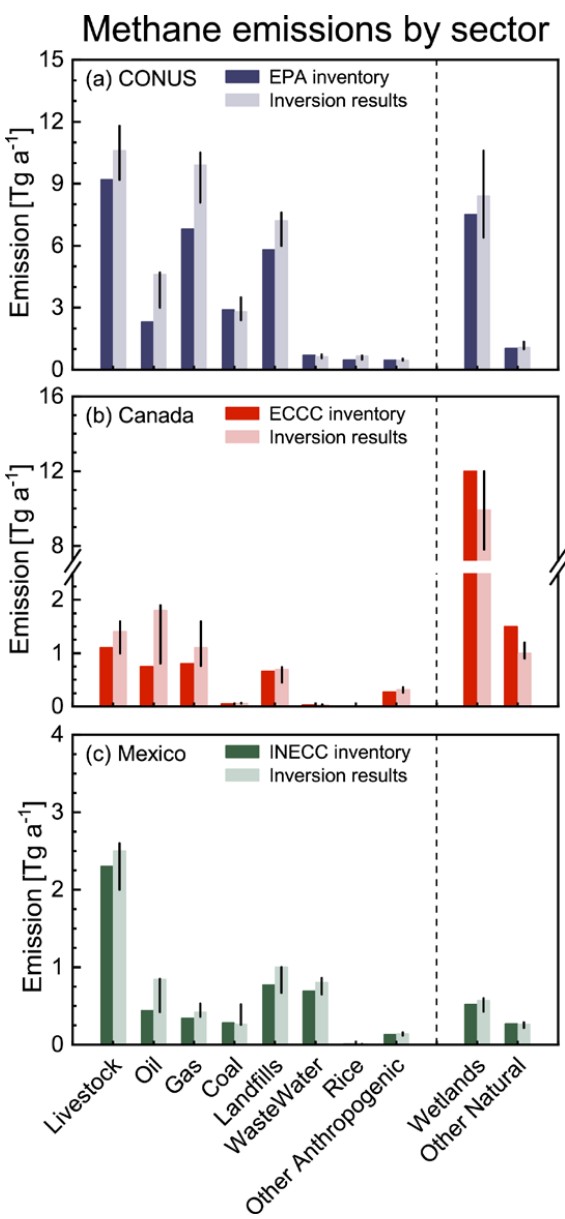

**Figure 7.** Mean 2010–2017 methane emissions by source sectors for the contiguous US (CONUS), Canada, and Mexico, displaying values and ranges from the corresponding Tables 1a-c. The official UNFCCC-reported national inventories for the US (EPA), Canada (ECCC) and Mexico (INECC) are used as prior estimates for the inversion. Inversion results are from the base inversion of GOSAT + in situ observations and the ranges are for the ensemble of 66 sensitivity inversions (Table 2) and sectoral attribution methods (Section 2.6). The dashed line separates anthropogenic and natural sources. Note the break in scale for Canada to accommodate the large wetlands source.

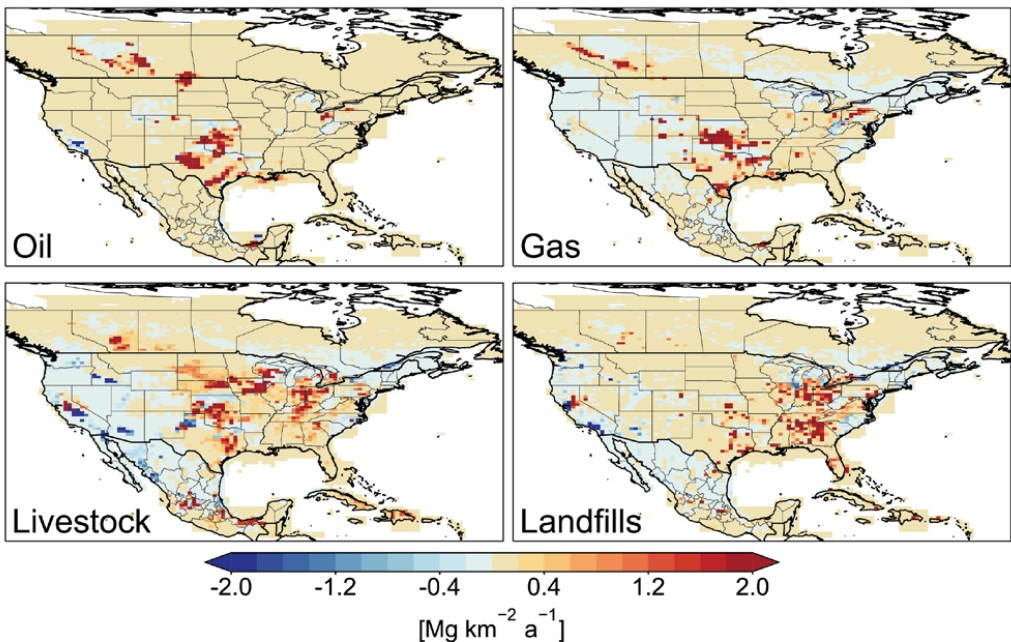

**Figure 8.** Posterior correction for mean 2010-2017 methane emissions from the oil, gas, livestock, and landfill sectors as given by the base inversion. The correction factors for anthropogenic emissions are relative to the national emission inventories used as prior estimates (EPA GHGI for the US, ECCC NIR for Canada, INECC for Mexico).


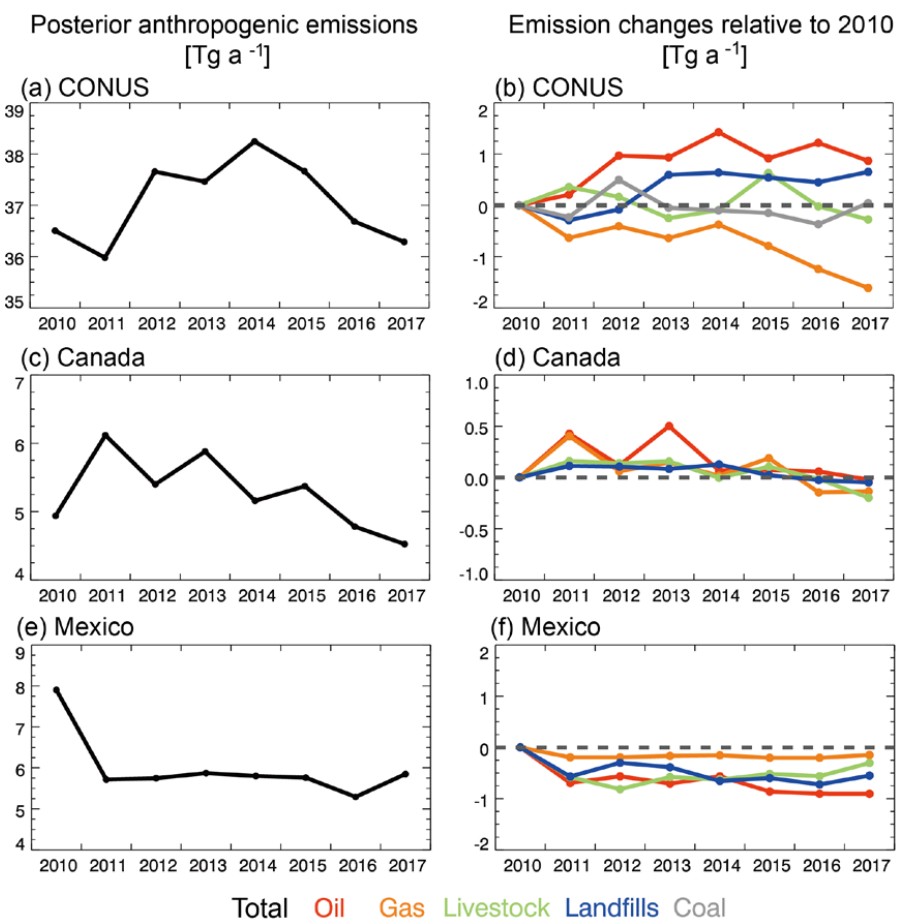

**Figure 9.** 2010-2017 trends in anthropogenic methane emissions in CONUS, Canada, and Mexico as inferred from the base inversion. The left panels show the total national anthropogenic methane emissions and the right panels show changes relative to 2010 for major sectors.

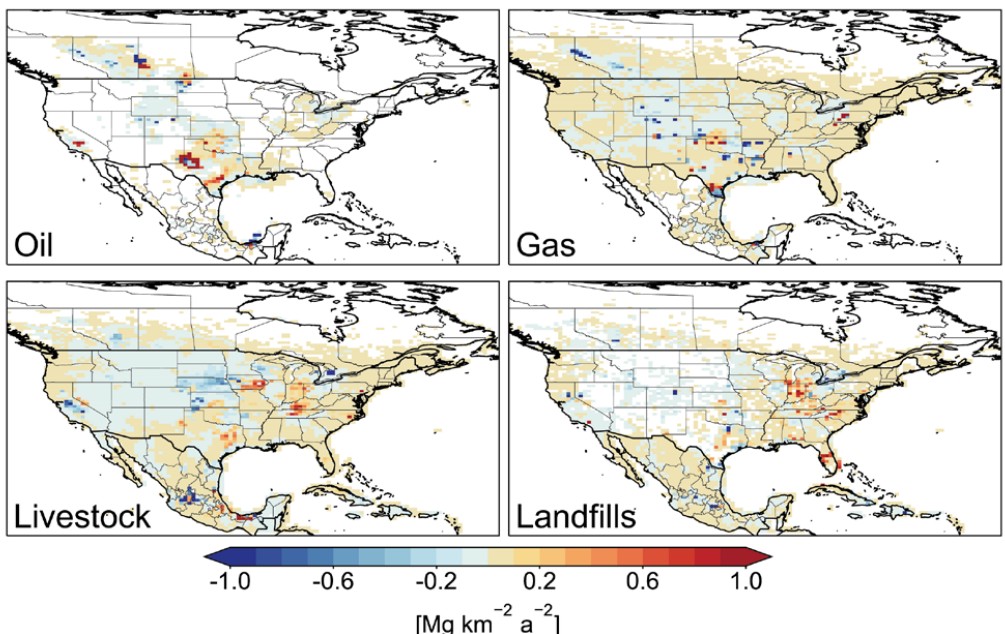

**Figure 10.** 2010-2017 linear trends in emissions from major anthropogenic sectors on the 0.5°x0.625° grid as inferred
from the base inversion. The linear trends are fitted by linear regression to the inversion results for individual years.
Areas in white have no emissions from the corresponding sector.

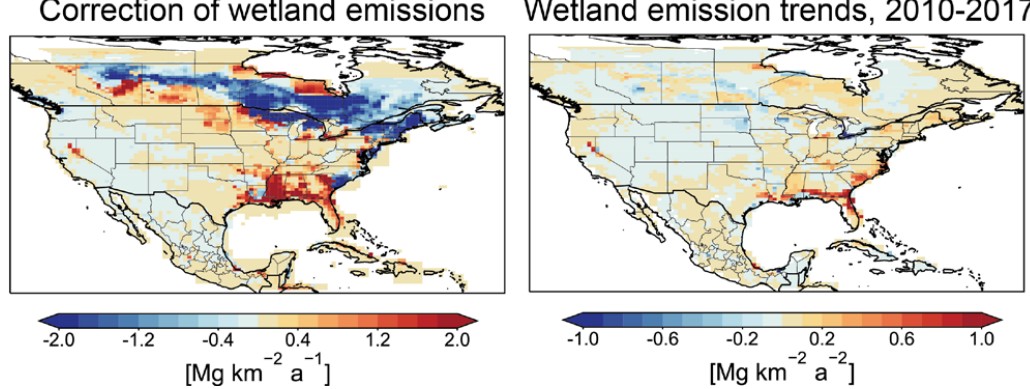

**Figure 11.** Posterior correction and linear trends for 2010-2017 wetland emissions in North America. The posterior correction factors are relative to the 2010-2017 mean of the high-performance subset of the WetCHARTs inventory ensemble (Ma et al., 2021). The linear trends are from ordinary linear regression to base inversion results for individual years.







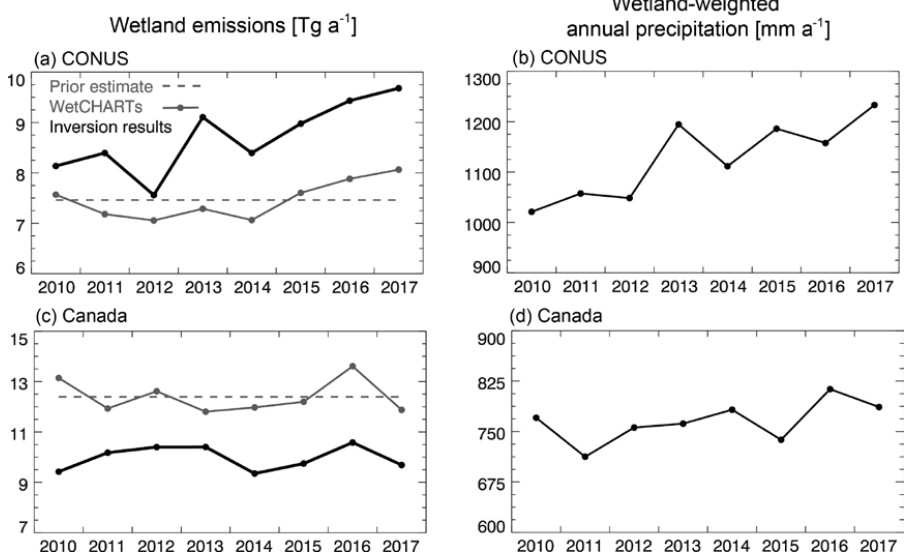

**Figure 12.** 2010-2017 trends of wetland methane emissions and precipitation in CONUS and Canada. The left panels show the results from the base inversion and the mean annual emissions from the high-performance ensemble of the WetCHARTs v1.3.1 inventory (Ma et al., 2021), The 8-year WetCHARTs average is used as prior estimate for the inversion so that the trend in the inversion results is solely from the atmospheric observations. The right panels show the annual precipitation over the wetland regions of CONUS and Canada, as determined by weighting precipitation amounts with the WetCHARTs wetland emission fluxes on their native 0.5ºx0.5º grid. The gridded precipitation data are from the ERA-Interim re-analyses (https://www.ecmwf.int/en/forecasts/datasets/reanalysis-datasets/era-interim/, 0.5°×0.5°).

