# Peer review of "Methane emissions in the United States, Canada, and Mexico: Evaluation of national methane emission inventories and sectoral trends by inverse analysis of in situ (GLOBALVIEWplus CH4 ObsPack) and satellite (GOSAT) atmospheric observations"

_Atmospheric Chemistry and Physics, 2021_

## Author Comment (AC1)

**Reviewer #1 Lena Höglund-Isaksson**

**Comment [1-1]:** GENERAL: I find this paper very interesting because it manages to shed considerable additional light on many of the big questions about the discrepancy between bottom-up and top-down estimates of methane emissions from North American sources. Without being an expert on inverse modelling myself (but rather bottom-up modelling), I still note that the authors make important improvements in the methodology that are additional to previous studies, i.e., using both satellite and in-situ observations, using a log-normal error function which better represents the high tail emission distributions that are typical for the oil and gas sources, and using an improved prior for wetlands, which does not overstate wetland emissions as has previously been a problem. These improvements seem to lead to results that better explain the total contribution from anthropogenic sources and their attribution to individual source sectors. The paper is also well written and easy to follow and I support publication but would like to see one major concern addressed and a few minor revisions, as listed below.

**Response [1-1]: We thank Dr. Lena Höglund-Isaksson for the positive and valuable comments. All of them have been implemented in the revised manuscript. Please see our itemized responses below.**

**Comment [1-2]:** MAJOR CONCERN: Authors are able to show that anthropogenic $CH_4$ emissions are substantially underreported in all three countries USA, Canada, Mexico, and in particular for the US. They conclude that in particular emissions from oil production are underreported by a factor of 2. Looking at the trend 2010-2017 for the US, they conclude that $CH_4$ emissions appear to have peaked in 2014 and thereafter slightly declined with the overall trend for the period still slightly increasing. This is in contrast to US official reporting to the UNFCCC, where emissions decline steadily over the period. The authors attribute the increases they find to oil production and landfill, while emissions from gas production are said to decline (and livestock and coal mining stay flat). Given that according to EIA, US shale gas production increased by 227% (from 165 to 540 bcm) over this period while oil production increased by a more modest 71% (and other natural gas production declined by 39% from 493 to 300 bcm), I am not convinced about the authors' split in attribution between oil and gas sector emissions. I wonder if the inversions can really make this distinction between oil and gas sources as fields in the US are often producing both oil and gas? If authors are not able to do this split in a robust manner, then I would recommend the authors not to report oil and gas sector emissions separately, because from a policy point of view this matters a lot. If there is a risk that authors are wrong about their conclusions here and that in reality it is a strong increase in methane emissions from shale gas production that is picked up (and not oil), then you risk sending the completely wrong signal to policy-makers (i.e., "fix oil but don't' worry too much about gas production", when it could be that the real problem is the shale gas). So if there is uncertainty regarding this, then report oil and gas emissions together and leave it to further research to figure out this split in more detail.

**Response [1-2]: Thank you for pointing it out. We have consulted with our EPA collaborators, on this important issue. They recommend separating oil and gas methane emissions in our inversion. This is because the national inventory is required to separate oil and gas emissions when reporting to the UNFCCC, but each country can make its own distinction between oil and gas so definitions will not always align. As our inversion uses the gridded inventory**

developed from the official GHGIs as the prior emissions, separating the oil and gas emissions in the posterior emissions can help focus areas of improvement on oil versus gas, which can be useful for compilers of the national reports. We prefer to follow their recommendation and keep the separation of oil and gas methane emissions.

We acknowledge that our methods for separating sectoral methane emissions may not be able to fully split the oil and gas emissions. We quantify this ability at the country-level by calculating the error correlation coefficients (r) in the inversion results for sector pairs, as show in Fig.6. We add the following statement in Section 3.2:"Optimization of the oil/gas sector is well separated from the other sectors in all three countries, and separation between oil and gas is also successful because the two sectors have very different spatial distributions in the gridded inventories (Figure 2). However, there is some ambiguity for the production subsectors, because wells often produce both oil and gas (Maasakkers et al., 2016), and for this reason some studies prefer to refer to oil/gas emissions as a combined sector (Alvarez et al., 2018). Separating oil and gas emissions is useful for our purpose because such separation is required under UNFCCC reporting, but the reader should be aware that this separation is done on the basis of the spatial distributions of emissions in Figure 2."

As for the EPA reports on sectoral emission trends, we find that EPA GHGI also estimates decreasing emissions from natural gas systems from 2010 to 2017 (EPA, 2021). This decrease is mainly driven by exploration (80% decrease from 2010 level) and distribution (12% decrease from 2010 level), while the emissions from gas production are quite flat, even though gas production has increased. The contributing factors to the decreasing gas emissions in the inversion results would require further analyses at basin and process level that we are now addressing in follow-up work.

We add the following statement in Section 3.3: "The EPA inventory reports no significant trend for oil emissions, and attributes the decrease in gas emissions to gas exploration (80% decrease from 2010 level) and distribution (12% decrease from 2010 level), with flat emission from gas production. However, both oil and natural gas productions have increased significantly over the period (https://www.eia.gov/). More work is required to understand the discrepancies in oil and gas trend estimates between the inversion and EPA reports. We cannot exclude the possibility that oil and gas emissions are not adequately separated in the EPA inventory and/or the inversion at this stage."

**Comment [1-3]:** MINOR CONCERN/EDIT: p.11 row 387: write out the acronym DOFS.
**Response [1-3]:** We have stated in Line 252 where "DOFS" were first used: "We refer to the diagonal elements of A as the averaging kernel sensitivities, and to the trace of A as the degrees of freedom for signal (DOFS), representing …". We also rephrase here "The number of independent pieces of information afforded by the observations (DOFS = 114) can be placed in the context of the 600 Gaussian state vector elements used to optimize the spatial distribution of emissions."

**Comment [1-4]:** p.14 row 530: It is suggested that the downward correction for offshore operations can be referred to that methane from offshore oil platforms is piped onshore and inefficiently flared. Another possible explanation could be that when methane leaks happen at the seabed, methane oxidises to CO2 in the water column before reaching the surface and therefore emissions are

considerably lower during offshore production. Could this be an explanation here?

**Response [1-4]: Thanks for bringing up this assumption. We have modified the text as "This is consistent with aircraft and TROPOMI satellite observations, which attributed the low offshore emissions to piping of the gas onshore followed by inefficient flaring (Zavala-Araiza et al., 2021; Shen et al., 2021). In addition, methane released to the ocean could be oxidized to CO2 in the oxic water and hence not reach the atmosphere."**

**Reference**

Shen, L., Zavala-Araiza, D., Gautam, R., Omara, M., Scarpelli, T., Sheng, J., Sulprizio, M. P., Zhuang, J., Zhang, Y., Qu, Z., Lu, X., Hamburg, S. P., and Jacob, D. J.: Unravelling a large methane emission discrepancy in Mexico using satellite observations, Remote Sens. Environ., 260, 112461, http://doi.org/10.1016/j.rse.2021.112461, 2021.

Zavala-Araiza, D., Omara, M., Gautam, R., Smith, M. L., Pandey, S., Aben, I., Almanza-Veloz, V., Conley, S., Houweling, S., Kort, E. A., Maasakkers, J. D., Molina, L. T., Pusuluri, A., Scarpelli, T., Schwietzke, S., Shen, L., Zavala, M., and Hamburg, S. P.: A tale of two regions: methane emissions from oil and gas production in offshore/onshore Mexico, Environmental Research Letters, 16, 024019, http://doi.org/10.1088/1748-9326/abceeb, 2021.

---

## Author Comment (AC2)

**Reviewer #2**

**Comment [2-1]: General comments:** The authors extend their previous coarse-grid global inversions to a fine-grid regional scale. They optimize methane emissions and 2010-2017 emission trends in North America by in situ (GLOBALVIEWplus CH₄ ObsPack) and satellite (GOSAT) observations, through analytical inversions using log-normal error forms. They point out large emission underestimates in the oil sector by a factor of 2, and a peak of CONUS anthropogenic emissions in 2014. The paper is well written. The methods are clearly described, and the results are well discussed. I support publication, but with a major concern and some minor suggestions.

**Response [2-1]: We thank the reviewer for the positive and valuable comments. All of them have been implemented in the revised manuscript. Please see our itemized responses below.**

**Comment [2-2]:** My major concern is that this study lacks independent evaluation. The authors compare the posterior simulation and prior simulation against the observations used for the inversions, and the improvements against GOSAT are weak. I am curious about the evaluation against the independent dataset, such as TCCON or other local in-situ measurements.

**Response [2-2]: Thank you for pointing it out. We have added independent evaluation with the five TCCON sites in US. We now state in the text: "Independent evaluation with the ground-based column observations from the Total Carbon Column Observing Network (TCCON) (Wunch et al., 2011) further shows that the mean model bias at five sites in CONUS decreases from 5.2-14.0 ppbv in the prior simulation to 1.0-13.5 ppbv in the posterior simulation."**

**Reference**

Wunch, D., Toon, G. C., Blavier, J.-F. L., Washenfelder, R. A., Notholt, J., Connor, B. J., Griffith, D. W. T., Sherlock, V., and Wennberg, P. O.: The Total Carbon Column Observing Network, Philos. T. R. Soc. A, 369, 2087–2112, https://doi.org/10.1098/rsta.2010.0240, 2011.

**Comment [2-3]: Specific comments and technical corrections.** Row 199: What are the treatments for the initial conditions of the global simulations?

**Response [2-3]: We now state in the text: "The initial methane concentration fields on 1 January 2010 are from Lu et al. (2021) which have been adjusted to have an unbiased zonal mean relative to GOSAT observations, such that model discrepancies with observations over our 2010-2017 simulation period can be attributed to model errors in emissions instead of errors in initial conditions."**

**Reference**

Lu, X., Jacob, D. J., Zhang, Y., Maasakkers, J. D., Sulprizio, M. P., Shen, L., Qu, Z., Scarpelli, T. R., Nesser, H., Yantosca, R. M., Sheng, J., Andrews, A., Parker, R. J., Boesch, H., Bloom, A. A., and Ma, S.: Global methane budget and trend, 2010–2017: complementarity of inverse analyses using in situ (GLOBALVIEWplus CH₄ ObsPack) and satellite (GOSAT) observations, Atmos. Chem. Phys., 21, 4637-4657, http://doi.org/10.5194/acp-21-4637-2021, 2021.

**Comment [2-4]:** Row 254: Are the Jacobian matrix for the boundary conditions constructed in the same way as the grid-level emissions?

**Response [2-4]:** Yes. We have clarified in the text: "We construct $K$ by conducting GEOS-Chem simulations where each element of the state vector (methane emission and model boundary correction) is perturbed separately."

**Comment [2-5]:** Row 297: The error standard deviations for boundary conditions are 10 ppb in the base inversion and 5 ppb in the sensitivity inversions. These are much smaller than the error standard deviations for emissions. How sensitive are the results if applying a larger error standard deviation for boundary conditions?

**Response [2-5]: We find that compared to the default 10 ppb used in the base inversion, using 5 ppb as the error standard deviations for boundary conditions leads to less than 1% difference in the resulting methane emissions in US, Canada, and Mexico, indicating its small sensitivity to the main conclusion. We did not use a larger error standard deviation because we have much more confidence in the boundary conditions than emissions, as it is generated from the global simulation using methane emissions and sinks optimized by GOSAT observations in our previous work (Lu et al., 2021). These global methane fields are unbiased to global zonal mean observations for 2010-2017, with some residual bias for individual years but even those are less 5 ppbv for the northern mid-latitudes (Fig.6 in Lu et al., 2021).**

**We now state in the revised text "Lu et al. (2021) show that their optimized simulation is unbiased in comparison to global zonal mean observations for 2010-2017 but we still find some residual biases for individual years up to 5 ppbv.".**

**Comment [2-6]:** Row 317: The observation error standard deviations for in-situ data are ~2× of that for GOSAT, and the total number of observations for in-situ data is 0.4× of that for GOSAT. Readers may be curious about the results if applying two regulation parameters separately to in-situ and GOSAT data.

**Response [2-6]: Yes, we indeed determine and apply two regulation parameters separately to the in-situ and GOSAT data. We now clarify in the text: "Here we determine the regularization factor $\gamma$ separately for in-situ and GOSAT data following Lu et al. (2021), and find that $\gamma = 1$ is best for the both. We also conduct a sensitivity inversion using $\gamma = 0.5$ for the GOSAT observation terms (while keeping $\gamma = 1$ for in-situ data terms in the joint inversion) as adopted in Maasakkers et al. (2021)."**

**Comment [2-7]:** Row 417: "This may reflect the underestimation of $CO_2$ over the Los Angeles Basin". Typo, $CH_4$ rather than $CO_2$?

**Response [2-7]: This is indeed the model $CO_2$ used in the proxy GOSAT retrieval, rather than $CH_4$. We now clarify in the text: "This may reflect the coarse resolution of model $CO_2$ used in the proxy GOSAT retrieval that leads to underestimation of $CO_2$ (and hence methane) over the Los Angeles Basin (Turner et al., 2015; Maasakkers et al., 2021), and/or complex topography."**

**Reference:**

Turner, A. J., Jacob, D. J., Wecht, K. J., Maasakkers, J. D., Lundgren, E., Andrews, A. E., Biraud, S. C., Boesch, H., Bowman, K. W., Deutscher, N. M., Dubey, M. K., Griffith, D. W. T., Hase, F., Kuze, A., Notholt, J., Ohyama, H., Parker, R., Payne, V. H., Sussmann, R., Sweeney, C.,

Velazco, V. A., Warneke, T., Wennberg, P. O., and Wunch, D.: Estimating global and North American methane emissions with high spatial resolution using GOSAT satellite data, Atmos. Chem. Phys., 15, 7049-7069, http://doi.org/10.5194/acp-15-7049-2015, 2015.

Maasakkers, J. D., Jacob, D. J., Sulprizio, M. P., Scarpelli, T. R., Nesser, H., Sheng, J., Zhang, Y., Lu, X., Bloom, A. A., Bowman, K. W., Worden, J. R., and Parker, R. J.: 2010–2015 North American methane emissions, sectoral contributions, and trends: a high-resolution inversion of GOSAT observations of atmospheric methane, Atmos. Chem. Phys., 21, 4339-4356, http://doi.org/10.5194/acp-21-4339-2021, 2021.

**Comment [2-8]:** Row 425: "For GOSAT the improvement is less apparent from the comparison statistics, because the prior simulation already has a low mean bias MB = -0.5 ppb, and the prior RMSE is only 6.9 ppb (which decreases to 6.5 ppb). However, we see from Figure 5 a significant whitening of the noise with reduction of regional-scale biases." This sounds like to contradict itself. It first presents that the regional mean bias of the posterior simulations is 0.6 ppb, larger than the -0.5 ppb in the prior simulations, then indicates that the posterior simulations' regional-scale bias is less than prior simulations in the map.

**Response [2-8]: Thanks for pointing it out. We have revised the text accordingly: "For GOSAT the improvement is less apparent from the continental-scale comparison statistics, because the prior simulation already has a low mean bias MB = -0.5 ppb and a small RMSE of 6.9 ppb. However, we see from Figure 5 that the small mean bias reflects an offset between high bias in western US and Canada and low bias in the central and eastern US. The inversion results in spatial whitening of this bias." We have also added independent evaluation against TCCON sites (Response [2-2]).**

---

## Author Response (AR2)

**acp-2021-671**

As of: Nov 12, 2021 9:26:07 AM
16,867 words - 127 matches - 6 sources

**Similarity Index**

**17%**

Mode: [ Similarity Report ⌄ ]
* * *
**paper text:**

[revised manuscript text omitted]

**by minimizing the Bayesian cost function**   $J_i$?($x_i$?) (   **Brasseur and Jacob, 2017):** $J_i$?($x_i$?) = ($x_i$? − $x_i$ ?$A_i$?)$^{T_i}$? $S_i$ ?$A_i$?−1( $x_i$? − $x_i$ ?$A_i$?) + $\gamma_i$?($y_i$ ? − $F_i$?( $x_i$ ?)$^{T_i}$? $S_i$ ?$O_i$?−1( $y_i$ ? − $F_i$?( $x_i$   | 6 |

?)) (1), where $x_i$?$A_i$? is the prior estimate of $x_i$?, $S_i$?$A_i$? denotes the

**prior error covariance matrix**   , $y_i$?   **is the observation vector**   , $S_i$?$O_i$? denotes   **the**   | 1 |

**observation error covariance matrix**   , $\gamma_i$?   **is a regularization factor**   | 1 |

(see below), and $F_i$?($x_i$?) represents the GEOS-Chem simulation of $y_i$?. The

**GEOS-Chem forward model**   $F_i$?($x_i$?)   **as implemented here is strictly linear**   | 3 |

(because methane sinks are not optimized), so that the model can expressed as $y_i$? = $K_i$?$K_i$? + $c_i$?, where $K_i$? = ∂i?$y_i$?/∂i?$x_i$?

**represents the Jacobian matrix and**   $c_i$?   **is**   a   **constant**   . Minimizing   **the**   | 1 |

cost function (Eq.1) by solving ∇i?$x_i$? $J_i$?($x_i$?) = 0 yields closed-form posterior estimates of the state vector ?$x_i$?, its error covariance matrix ?$S_i$?, and the averaging kernel matrix $A_i$? (Rodgers, 2000; Brasseur and Jacob, 2017): $x_i$?? = $x_i$?$A_i$? + $G_i$?(

$$\boldsymbol{y}\text{i? } - \boldsymbol{K}\text{i?}\boldsymbol{x}\text{i} \quad \text{?}\boldsymbol{A}\text{i?}) \text{ (2)}, \quad \boldsymbol{S}\text{i?? } = (\gamma\text{i?}\boldsymbol{K}\text{i} \quad \text{?}\boldsymbol{T}\text{i?} \quad \boldsymbol{S}\text{i} \quad \text{?}\boldsymbol{O}\text{i?}{-1}\boldsymbol{K}\text{i? } + \quad \boldsymbol{S}\text{i}$$

?$A$i?−1)−1 (3), $A$i? = ∂i?$x$i?? = $I$i?$n$i? − ?$S$i?$S$i?$A$i?−1i? (4), ∂i?$x$i? where $G$i? in Eq.2 is the gain matrix, $G$i? = ∂i?$x$i?? = ($\gamma$i? $K$i?$T$i?$S$i?$O$i?−1$K$i? + $S$i?$A$i?−1)−1$\gamma$i?$K$i?$T$i?$S$i?$O$i?−1 (5). ∂i?$y$i? 250 The averaging kernel matrix A in Eq. 4 quantifies the

**sensitivity of the posterior estimate to**    changes in    **the true**

[revised manuscript text omitted]

$s_A$) following (Maasakkers et al., 2019): 2 s'A = ? $(\ln x_A + s_A + \ln x_A - s_A) x_A x_A$ 2 ? (7). We adopt as convergence criterion that the maximum difference between $x'_{N+1}$ and $x'_N$ elements be smaller than 5‰, at which point we adopt $x'' = x'_{N+1}$ as our posterior solution. The posterior error covariance and averaging kernel matrices $S''$ and A' on the log solution are obtained by replacing $S_A$ 300 and $K$ with $S'_A$ and $K'$ in Eqs. (3) and (4). Optimization of

**emissions in log space means that** $?x'$ is a **best estimate** of **the median of the** | 5 |

log-normal error distribution rather than the mean. The mean values for spatial and sectoral aggregation purposes can be inferred from the properties of the lognormal distribution as $x_{ji}(m_i m_i m_i) = x_{ji}(m_i m_i m_i m_i) e^{s''_{ji}/2}$ where $s''_{ji}$ is the

**corresponding diagonal element of the** 8 305 posterior **error covariance matrix** | 3 |

in log space, i.e., the geometric error standard deviation. The boundary conditions are still optimized with normal error distributions, assuming an error standard deviation of 10 ppb. 310 The above describes our base inversion. We also conduct sensitivity inversions using different error assumptions. This includes 1) using the quadrature sum of

**error variances for all sectors contributing to a given Gaussian** | 3 |

with a cap of 50% following Maasakkers et al. (2021), resulting in a 43% error on average; 2) to 4) using the normal error distributions (then with the linear Jacobian matrix) with 50%, 95%, and the quadrature sum of errors for individual Gaussians as error variances; 5) assuming an error standard deviation of 5 ppb for boundary conditions. 315 The observation

**error covariance matrix** $S_O$ **includes contributions from** measurement **and forward model errors. We** compute **it** following **the residual error method** originally described by **Heald et al. (2004** | 5 |

) and previously used by Lu et al. (2021). A GEOS-Chem simulation with prior emission estimates yields a prior model estimate $F(x_A)$ of concentrations at the observation points. The mean 2010-2017 discrepancy between the

observations and the prior model, $?y_i??? - ??? F_i???(? x_i ??? A_i ???)$, is determined for each grid cell (for GOSAT), individual observation site (surface and tower), and observation platform (shipboard and aircraft). $y_i???? - ??? F_i???(? x_i ??? A_i ??)?$ is taken to represent the systematic bias in the prior emissions to be corrected in the inversion. The residual term,

$$\varepsilon_i?O_i? = y_i? - F_i?(x_i?A_i?) - ?y_i??? - ??? F_i???(?x_i???A_i ???), \quad \text{represents} \quad \textbf{the} \qquad \boxed{6}$$

random observation error including contributions from the measurements, the forward model, and the representation of the observation points on the model grid (Heald et al., 2004). The variance of $\varepsilon_i?O_i?$ provides the diagonal terms of $S_i?O_i?$. The resulting observation error standard deviations average 13 ppb for GOSAT, 26 ppb for surface sites, 39 ppb for towers, 19 ppb for ships, and 22 ppb for aircraft. The observation error is larger for in situ than for satellite observations, even though the in situ measurements are more precise, because the forward model error is larger for vertically resolved points (particularly for surface air in source regions) than for atmospheric columns (Cusworth et al., 2018). The

**observation error for in situ observations is dominated by the forward model error**     $\boxed{1}$

while that for GOSAT is dominated by the measurement error. We do not have sufficient objective information to quantify the error correlation structure of SO and we therefore assume it to be diagonal. This may underestimate $S_i?O_i?$ because of correlated

**transport and source aggregation errors in the forward model**     $\boxed{1}$

, as noted above. We follow Zhang et al. (2018) to introduce a regularization factor $\gamma_i?$ for the observation terms in the cost function $J_i?(x_i?)$ (Eq. 1) to avoid either overfits or underfits that would result from missing covariant (off-diagonal) structure in $S_i?O_i?$ and $S_i?A_i?$, respectively. Lu et al. (2021) showed that the optimal value of this regularization factor can be selected such that the sum of the n prior

**terms in the posterior**    estimate    **of the cost function**     $\boxed{5}$

$(J_i?A_i?(?x_i?) = (?x_i? - x_i?A_i?)T_i?S_i?A_i?-1_i?(?x_i? - x_i?A_i?))$ has a value $\approx$ n, which is the expected value from the

**Chi-square distribution with n degrees of freedom**     $\boxed{1}$

. Here we determine the regularization factor $\gamma_i?$ separately for in-situ and GOSAT data following Lu et al. (2021), and find that γ = 1 is best for both.

**We also conduct** a **sensitivity** inversion **using γ = 0.5 for** [1]

the GOSAT observation terms (while keeping γ = 1 for in-situ data terms in the joint inversion) as adopted in Maasakkers et al. (2021). Table 2 summarizes the settings of our base inversion (in bold) and the inversion ensemble. The ensemble comprises 33 inversions using the different combinations of settings in the Table. The base inversion including GOSAT + in situ data represents our best estimate, but we will compare it prominently to the

**GOSAT-only and in-situ−only inversions** with **the** same inversion parameters **in** [1]

order to evaluate the contributions from the different observing platforms for optimizing emissions. We will use the other ensemble members to discuss the sensitivity of inversion results to the choices of observations and inversion parameters, and to define the range of uncertainty in the inversion results. 2.6 Sectoral attribution and aggregation of inversion results The inversion returns the posterior estimates of mean emissions for each of the Gaussians, and we allocate these emissions to the native 0.5o×0.625o model grid by summing the contributions of all Gaussians on the grid. This defines a correction factor f0 to total prior emissions for each 0.5o × 0.625o grid cell and including the contributions from all q emission sectors (in our case q = 12, cf. Table 1). For sectoral attribution of this total correction factor we need to derive the correction factors fi to the individual sectors $ii? \in [1, qi?]$ contributing to f0. We use two alternative methods for this purpose. The first method simply takes fi = f0 for all i, thus assuming that the partitioning of sectoral

**emissions in individual grid cells is correct in the prior inventory** [1]

and all sectors contribute equally to the grid-level correction factor (Maasakkers et al., 2021; Lu et al., 2021; Zhang et al., 2021) .

**These assumptions are** reasonable **when the sectors are spatially separated, but** [1]

may be source of error when they spatially overlap. The second method (Shen et al., 2021) accounts for emissions from different sectors having different prior error standard deviation σi and therefore contributing differently to f0. Following Shen et al. (2021), $fi?ii?$ is then given by: $fi?ii? = 1i? - \eta\alpha i?ii?\sigma i?2i?ii?\sigma i?,(2i?Ai?1i?-fi?0i?),$ (8) where $\alpha i?ii?$ is the fraction of total emissions in the grid cell contributed by sector i, $\sigma i?Ai?$ is the prior error standard deviation for total emissions in the grid cell, and η = $qi? \sigma i?Ai?2$ 2 is a normalization factor. For the 2 $\sum ii?=1 \alpha i?ii? \sigma i?ii?$ prior error standard deviations σi on the 0.5o×0.625o grid

> **we use the scale-dependent** adaptation by **Maasakkers et al. (2016) of**    [5]

EPA sectoral national error estimates. This results in prior error standard deviations of 43% for rice, 66% for wastewater, 51% for landfills, 38% for livestock, 18% for coal, 30% for gas, and 87% for oil emissions. We further use 70% for wetlands (Bloom et al., 2017) and 100% for all other natural sources. These error estimates are solely used to infer fi values in equation (8), so that more uncertain emissions will contribute more to the correction. We use the second method in our base attribution of posterior estimates to emission sectors but will also use the results from the first method to contribute to error ranges in these sector-attributed posterior estimates. We also need to aggregate posterior emission estimates nationally and by sector for comparison to the national emission inventories. Following Maasakkers et al. (2019), this is done by a

> **transformation from the posterior full-dimension state vector** $\hat{x}i?$ **to the reduced state vector**    [1]

$\hat{x}i?ri?ri?ri?$ (national emission for a given sector) with a summation matrix W: 385 390 $\hat{x}i?ri?ri?ri?$ = $W$i?? $\hat{x}i?$ (9). The

> **posterior error covariance and averaging kernel matrices for the reduced state vector** are **then**    [1]

given by $\hat{S}i?ri?ri?ri?$ = $W$i?? $\hat{S}i?W$i? $T$i? (10), $A$i?ri?ri?ri? =

> $W$i? $W$i? $W$i ?∗ (11), **where** $W$i?∗ = $W$i? $T$i?($W$i? $W$i? $T$i?)−1i    [6]

[revised manuscript text omitted]
/, 0.5°×0.5°). 1160 15 30 175 255 295 320 325 330 335 340 350 355 360 365 370 375 380 400 405 410 415 420 430 435 440 445 455 1080 1105 345 425 1 2 3 7 9 10 14 17 19 25 26 27 28 29 30 31 32 33 34 35 36 37 38 39 40 41 42

**sources:**

| 1 | 669 words / 5% - Internet from 03-May-2021 12:00AM |
|---|---|
| | acp.copernicus.org |

| 2 | 454 words / 4% - Internet from 20-Aug-2021 12:00AM |
|---|---|
| | www.sciencegate.app |

| 3 | 401 words / 3% - Internet from 17-Oct-2021 12:00AM |
|---|---|
| | acp.copernicus.org |

| 4 | 312 words / 2% - Internet from 27-Sep-2021 12:00AM |
|---|---|
| | www.researchgate.net |

| 5 | 180 words / 1% - Internet from 12-Nov-2021 12:00AM |
|---|---|
| | acp.copernicus.org |

 165 words / 1% - Internet from 21-May-2021 12:00AM
acp.copernicus.org